# A PPARγ transcriptional cascade directs adipose progenitor cell-niche interaction and niche expansion

Yuwei Jiang[1,2,*], Daniel C. Berry[1,2,*], Ayoung Jo[2], Wei Tang[2], Robert W. Arpke[3,4], Michael Kyba[3,5] & Jonathan M. Graff[1,2,6]

Adipose progenitor cells (APCs) reside in a vascular niche, located within the perivascular compartment of adipose tissue blood vessels. Yet, the signals and mechanisms that govern adipose vascular niche formation and APC niche interaction are unknown. Here we show that the assembly and maintenance of the adipose vascular niche is controlled by PPARγ acting within APCs. PPARγ triggers a molecular hierarchy that induces vascular sprouting, APC vessel niche affinity and APC vessel occupancy. Mechanistically, PPARγ transcriptionally activates PDGFRβ and VEGF. APC expression and activation of PDGFRβ promotes the recruitment and retention of APCs to the niche. Pharmacologically, targeting PDGFRβ disrupts APC niche contact thus blocking adipose tissue expansion. Moreover, enhanced APC expression of VEGF stimulates endothelial cell proliferation and expands the adipose niche. Consequently, APC niche communication and retention are boosted by VEGF thereby impairing adipogenesis. Our data indicate that APCs direct adipose tissue niche expansion via a PPARγ-initiated PDGFRβ and VEGF transcriptional axis.

[1] Division of Endocrinology, Department of Internal Medicine, University of Texas Southwestern Medical Center, Dallas, Texas 75390, USA. [2] Department of Developmental Biology, University of Texas Southwestern Medical Center, Dallas, Texas 75390, USA. [3] Lillehei Heart Institute, University of Minnesota, Minneapolis, Minnesota 55455, USA. [4] Department of Medicine, University of Minnesota, Minneapolis, Minnesota 55455, USA. [5] Department of Pediatrics, University of Minnesota, Minneapolis, Minnesota 55455, USA. [6] Department of Molecular Biology, University of Texas Southwestern Medical Center, Dallas, Texas 75390, USA. * These authors contributed equally to this work. Correspondence and requests for materials should be addressed to D.C.B. (email: daniel.berry@utsouthwestern.edu) or to J.M.G. (email: jon.graff@utsouthwestern.edu).

Adult stem cells typically reside in a specialized microenvironment termed the niche[1,2]. The niche is a central means through which stem cells are controlled and regulated[1,2]. Niches also have the potential for regenerative medicine, for example, in *ex vivo* applications to regulate stem cell transformations and to direct stem cell reprogramming[3,4]. Therefore, niches hold therapeutic potential to regulate stem cell action and tissue generation, repair and homeostasis and systemic metabolism[5,6]. Recently, several studies have indicated that stem cells can regulate their niche but the molecular and cellular understanding of how niches are formed and maintained, how stem cells localize to the niche and how stem cells are maintained in the niche remains unclear[7].

A stromal perivascular niche has been proposed for several stem compartments, including neural stem cells, hematopoietic stem cells and mesenchymal stem cells as well as adult adipose progenitor cells (APCs)[8–11]. In the adipose lineage, adult APCs are a subset of perivascular mural cells (for example, pericytes, vascular smooth muscle cells) that adhere to blood vessel walls and subsequently differentiate into adipocytes upon adipogenic cues[10–17]. Accordingly, the white adipose tissue (WAT) vasculature and stromal surrounding appears to serve as the APC niche[10,11,18]. Consistent with their mural character, adult APCs express a battery of classical mural markers and can be marked and manipulated with several mural marker-based genetic systems such as smooth muscle actin (SMA)-Cre$^{ERT2}$ and *SMA-rtTA*[10,12]. The SMA+ APC mural subset can be further identified because they express peroxisome proliferator-activated receptor gamma (PPARγ). PPARγ is a nuclear hormone receptor and a master transcriptional regulator of adipocyte differentiation[19–21]. These adult APCs acquire vascular residence after a distinct and independent developmental APC compartment forms the adipose depots during adipose tissue organogenesis; thus APCs are not born on blood vessels rather they migrate, adhere and then reside in perivascular positions[10,12,13,22]. Yet it is not understood how adult APCs acquire blood vessel niche occupancy and how they are maintained in the niche.

Here we found that transcriptional activation of PPARγ within APCs acts as a central regulator of WAT niche expansion. PPARγ controls niche expansion by stimulating APC niche interaction and retention and by increasing vasculogenesis via a transcriptional network that includes platelet-derived growth factor receptor beta (*Pdgfrβ*) and vascular endothelial growth factor (*Vegf*). Together, our studies suggest that adult APCs control the expansion and maintenance of their own niche through a PPARγ to PDGFRβ and PPARγ to VEGF molecular network, which can influence adiposity and systemic metabolism.

## Results

**PPARγ regulates APC niche occupancy.** PPARγ is expressed in the APC lineage before and during adipose tissue vascular niche assembly and APC niche occupancy[10]. To probe possible roles of PPARγ in adipose niche biology, we turned to PPARγ loss-of-function (LOF) and PPARγ gain-of-function (GOF) genetic strategies. We previously described two PPARγ LOF mouse models[10,23]. In one, a *Pparγ* conditional allele (*Pparγ*$^{fl}$) is deleted using a *Sox2-Cre* strain that expresses Cre in the epiblast (Fig. 1a). This approach generates a whole body *Pparγ* null, yet avoids the embryonic lethality of constitutive nulls that is secondary to roles of PPARγ in placental vascular function[18,24]. We also disrupted PPARγ in a restricted fashion throughout adipose lineage specification using AdipoTrak-Cre (*Pparγ*$^{+/tTA}$; *Tre-Cre*) (Fig. 1a). To track and visualize APC location and pattern, we incorporated the adipose lineage-tracking mouse model, with a

TRE-H2B-GFP allele (*Pparγ*$^{+/tTA}$; *Tre-Cre*; *Tre-H2B-GFP*) (Fig. 1a)[11]. We termed the ensuing strains Sox-PPARγ-LOF and AT-PPARγ-LOF (Fig. 1a). The two LOF strategies altered *Pparγ* expression in the intended manner based upon quantitative real-time PCR (qPCR) analyses of fluorescence-activated cell sorting (FACS)-isolated green fluorescent protein-positive (GFP+) progenitors and both mice were lipodystrophic as previously described (Fig. 1b and Supplementary Fig. 1a–c)[10,23].

The PPARγ-positive adult adipose stem lineage originates in a dorsal anterior position at approximately E10.5 (ref. 10). These cells then undergo a rostral–caudal migratory stream to enter adipose depots between postnatal (P) day P20 and P30 (ref. 10). To determine the requirement of PPARγ for APC migration, we examined whole-tissue explants for APC locality by GFP fluorescence and found that GFP+ APCs were present in adipose depots of 2-month-old controls and both LOF mutant strains (Fig. 1c). Higher magnification images of cryosections confirmed the presence of the progenitors in controls and LOF strains (Supplementary Fig. 1c)[10,23]. Together these studies indicate that PPARγ-deficient GFP+ APCs are present in the proper gross anatomical adipose depot locale.

We next examined whether PPARγ-LOF APCs occupied the perivascular niche. Immunostained sections for GFP (progenitor marker), CD31 (endothelial marker) and SMA (mural perivascular marker) demonstrated a reduction in APC–perivascular niche positions in PPARγ-LOF mice (Fig. 1d). Quantitating the distance of APCs away from adipose tissue CD31+ blood vessels confirmed our immunostaining results (Supplementary Fig. 1d). This lack of perivascular niche locality was also apparent in stromal vascular particulates (SVPs), an organotypic assay to maintain the native niche structure, isolated from either PPARγ-LOF model (Fig. 1e and Supplementary Fig. 1e).

To complement the PPARγ-LOF strategy and to test whether PPARγ could promote niche formation, we developed a PPARγ GOF (*Tre-Pparγ*) mouse strain[25]. We then combined the *Tre-Pparγ* allele with AdipoTrak-GFP (AT-GFP), generating AT-PPARγ-GOF mice. To determine whether the AT-PPARγ-GOF allele could rescue the AT-PPARγ-LOF phenotype, we ingressed the allele into the AT-PPARγ-LOF strain, generating AT-PPARγ-Rescue mice (Fig. 1a). The AT-PPARγ-GOF allele appeared functional based upon *Pparγ* expression levels, body fat analyses and molecular marker quantification and also upon rescue of the PPARγ-LOF lipodystrophic phenotype (Supplementary Fig. 1a–g). This notion is also supported by immunohistochemical and organotypic studies of the GOF and Rescue depots that showed an increase, compared to controls, in the number of progenitors located at the vascular niche (Fig. 1b–e and Supplementary Fig. 1d,e). These data support the notion that PPARγ, acting in the APCs, regulates APC niche localization.

**PPARγ regulates WAT vascular expansion.** We next attempted to elucidate whether PPARγ, acting within the APCs, regulated WAT vascular niche expansion. Towards this end, we scored the number of vessels in PPARγ-LOF and PPARγ-GOF depots by haematoxylin and eosin (H&E) staining, CD31 immunohistochemistry and CD31 quantification (Fig. 2a and Supplementary Fig. 1h,i). We found that PPARγ-LOF depots had fewer vessels, whereas PPARγ-GOF depots had many. Cell number and expression of CD31 and SMA cells using flow cytometry from the various mouse models expressing different levels of PPARγ confirmed these findings (Fig. 2b and Supplementary Fig. 1h). To attempt to understand the vascular phenotypes, we profiled mRNA expression of vasculogenic genes. Overall, we found reduced vasculogenic gene expression in SV cells extracted from

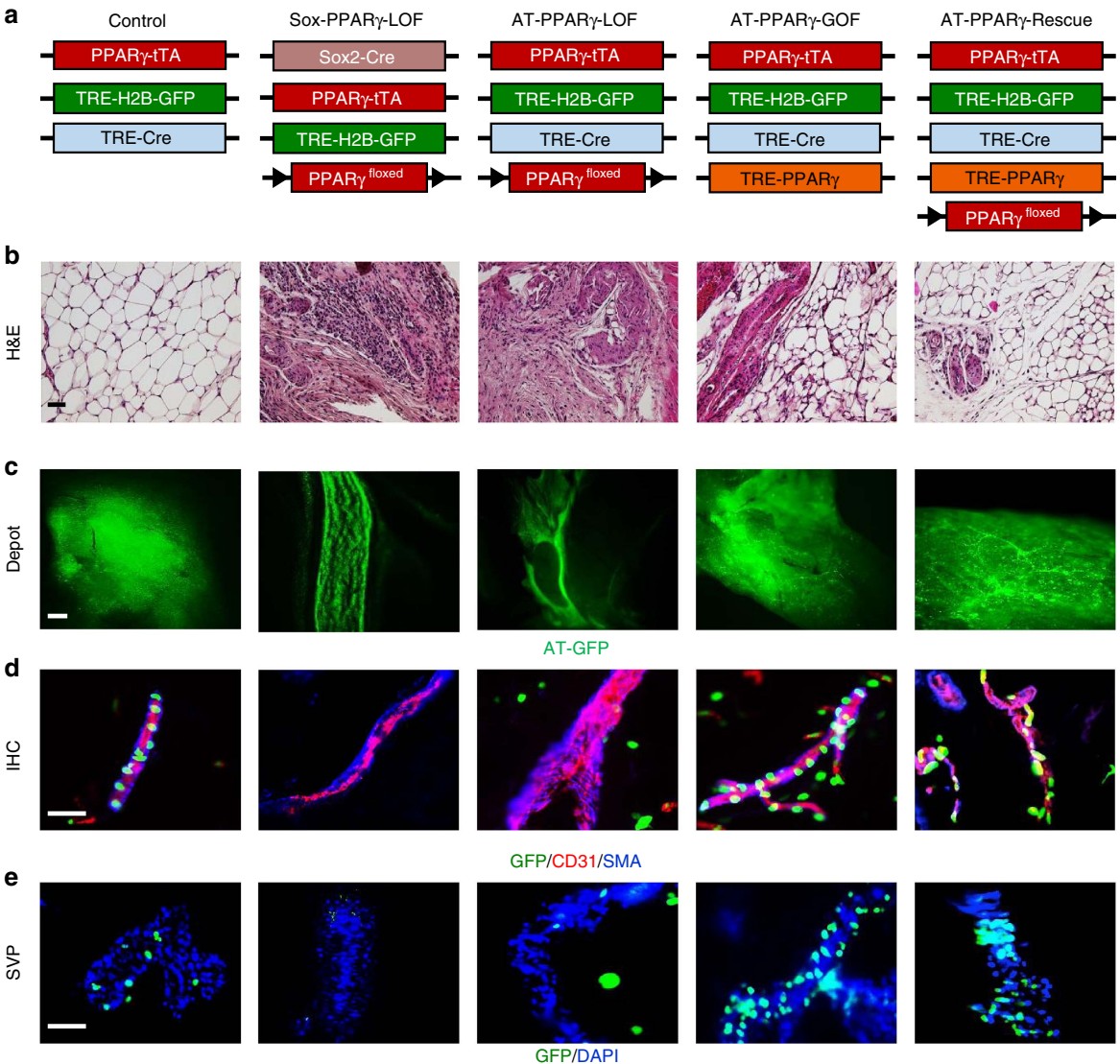

**Figure 1 | PPARγ regulates APC–blood vessel residency.** (**a**) Illustration of genetic alleles used to generate: AdipoTrak (AT)-control (*PPARγ^tTA*; *TRE-H2B-GFP*; *TRE-Cre*); Sox-PPARγ-LOF (*Sox2-Cre*; *PPARγ^f/tTA*; *TRE-H2B-GFP*); AT-PPARγ-LOF (*PPARγ^f/tTA*; *TRE-Cre*; *TRE-H2B-GFP*); AT-PPARγ-GOF (*PPARγ^tTA*; *TRE-H2B-GFP*; *TRE-PPARγ*); and AT-PPARγ-Rescue (*PPARγ^f/tTA*; *TRE-PPARγ*; *TRE-Cre*; *TRE-H2B-GFP*). Experiments were performed three times on six mice per group. (**b**) Representative H&E images from mice described in **a**. (**c**) Representative GFP images of freshly isolated subcutaneous IGW depots from mice described in **a**. Scale bar 10 mm. (**d**) Representative images of CD31 (endothelial marker) and SMA (mural cell marker) and APC-GFP immunostaining from subcutaneous IGW depots from mice described in **a**. (**e**) Representative images of SVPs from subcutaneous adipose depots from mice described in **a**. Locality of APCs was assessed by GFP fluorescence 12 h after isolation. DAPI was used to visual nuclei and cell number (*n* = 9). Scale bars 100 μm.

LOF adipose depots, whereas SV cells from PPARγ-GOF and AT-PPARγ-LOF-Rescue showed a reciprocal pattern (Fig. 2c).

To identify whether PPARγ might potentially regulate niche expansion, we used organotypic-vasculogenic approaches[10]. For this, we encased WAT explants in Matrigel and monitored vascular outgrowths and APC vessel occupancy. We found that both models of PPARγ-LOF depots were defective in vascular sprouting: they had fewer vascular sprouts, reduced sprout length, and a low number of branching points compared to controls (Fig. 2d–f). We also observed a reduction in GFP + progenitor vascular occupancy (GFP + progenitors/vascular sprout) in both LOF strains (Fig. 2g). In contrast, explants from PPARγ-GOF and PPARγ-LOF-Rescue had increased vascular sprouting, sprout length, sprout branching, and GFP + progenitor cell occupancy per sprout (Fig. 2d–g).

**PPARγ regulates WAT APC niche interaction and expansion.** Formation of adipocytes, during adipose organogenesis, is disrupted in the Sox2- and AT-PPARγ LOF strains, which could contribute to the vascular niche phenotypes observed above (Figs 1 and 2). To address this concern, we turned to the previously described mural-APC driver, *SMA-Cre^ERT2*, allowing for adult-specific APC interrogation in intact adipose tissues[10,12]. We incorporated the *Pparγ^fl* allele with the *SMA-Cre^ERT2* driver as well as the AT-GFP lineage tracking system to monitor the adult APCs (SMA-PPARγ-LOF) (Supplementary Fig. 2a). We induced deletion of PPARγ by administering tamoxifen (TM, 50 mg kg^−1) for 2 consecutive days (Supplementary Fig. 2b,c). Mice were then chased for 14 days before changes in fat content (Supplementary Fig. 2d)[10]. Histological examination revealed normal adipose tissue architecture, morphology, vascularity and

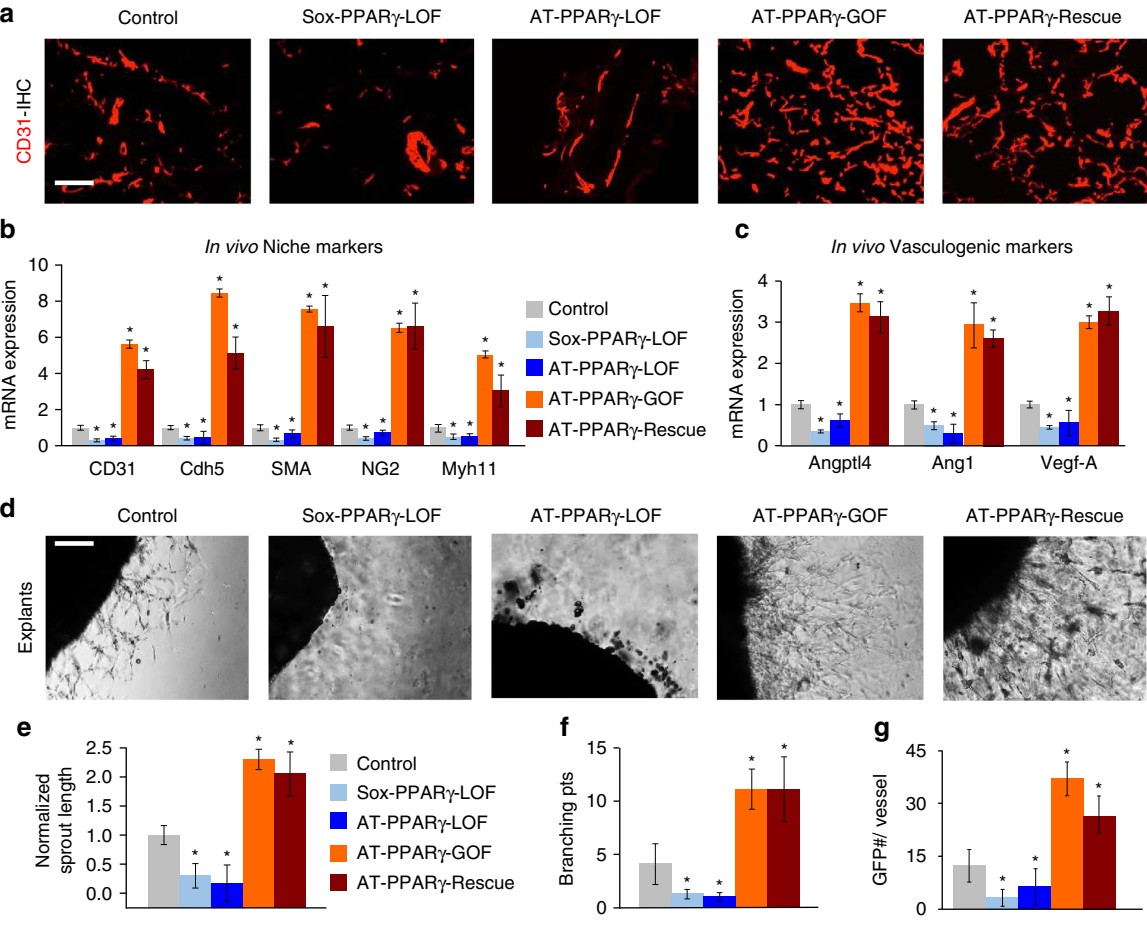

**Figure 2 | PPARγ is required for WAT niche expansion.** (**a**) Representative sections from AT-control, Sox-PPARγ-LOF, AT-PPARγ-LOF, AT-PPARγ-GOF and AT-PPARγ-Rescue stained for endothelial marker CD31. (**b**,**c**) Quantitative RT–PCR analysis of endothelial and mural cell (nichegenic) markers (**b**) and vasculogenic genes (**c**) from total SV cells isolated from the mice described in **a**. (**d**–**g**) Representative images of vascular sprouts of subcutaneous IGW explants from the mice described in **a**, **d**. Sprout length (**e**), branching points (**f**) and GFP progenitor cell occupancy (**g**) were quantified. Scale bars 100 μm. Data are means ± s.e.m. Experiments were performed three times on eight mice per group. *$P < 0.05$ mutant compared to control levels unpaired $t$-test, two-tailed.

mature adipocyte gene expression, in the mutant mice (Supplementary Fig. 2e,f). Similarly, flow cytometric and qPCR studies indicated the preservation of the overall vascular endothelial compartment in both cohorts of mice (Supplementary Fig. 2g,h)[10]. Markedly, deleting PPARγ in adult APCs reduced APC niche interaction as assessed by immunostaining and SVPs (Supplementary Fig. 2i–l). The APCs deficient in PPARγ appeared in clusters and further away from blood vessels (Supplementary Fig. 2k). In agreement, APC PPARγ-deficient WAT explants demonstrated diminished vascular sprouting, sprout length and branching and GFP+ progenitor cell occupancy per sprout (Supplementary Fig. 2m–p). Additionally, WAT vasculogenic genes were reduced in PPARγ-deficient animals (Supplementary Fig. 2q).

We next examined whether PPARγ was sufficient to stimulate niche expansion when expressed in adult APCs within the vascular niche compartment by generating *Sma-rtTA; Tre-Pparγ* mice (Supplementary Fig. 3a–c). However, this doxycycline (Dox)-On system precluded our ability to use the Dox-Off AT-GFP APC marking and therefore limited our APC niche occupancy analyses. Nonetheless, we found that PPARγ was sufficient to stimulate vessel formation *in vivo* and in *ex vivo* explant assays. PPARγ was also sufficient to induce vasculogenic gene expression in WAT depots (Supplementary Fig. 3d–g).

Taken together, these data indicate that PPARγ acts within the adult APC compartment to direct formation and maintenance of the adult niche.

We then tested whether transcriptionally activating PPARγ using rosiglitazone, a thiazolidinedione synthetic PPARγ agonist, could alter nichegenic potential and APC niche interaction[26,27]. Towards this end, we administered vehicle or rosiglitazone to AT-GFP lineage tracking mice via food for 2 weeks. Rosiglitazone treatment increased the vascularity of adipose depots based upon H&E, flow cytometry and qPCR analyses, similar to our PPARγ-GOF models (Supplementary Fig. 4a–d). This also appeared to be dependent on PPARγ transcriptional action; as a PPARγ antagonist, GW9662 (ref. 28) blocked the ability of rosiglitazone to induce vasculogenic gene expression *ex vivo* (Supplementary Fig. 4e). SVP studies indicated that rosiglitazone increased the number of APCs in the perivascular niche (Supplementary Fig. 4f,g). In WAT explant assays, rosiglitazone increased vascular sprouting, sprout length, sprout branching and GFP+ APC occupancy compared to vehicle (Supplementary Fig. 4h–k). Rosiglitazone also upregulated numerous vasculogenic genes and stimulated cellular migration (Supplementary Fig. 4l–n). Thus transcriptional activation of PPARγ increases WAT blood vessel niche expansion, as observed in humans[29], by increasing vasculogenic genes and by stimulating APC niche interaction.

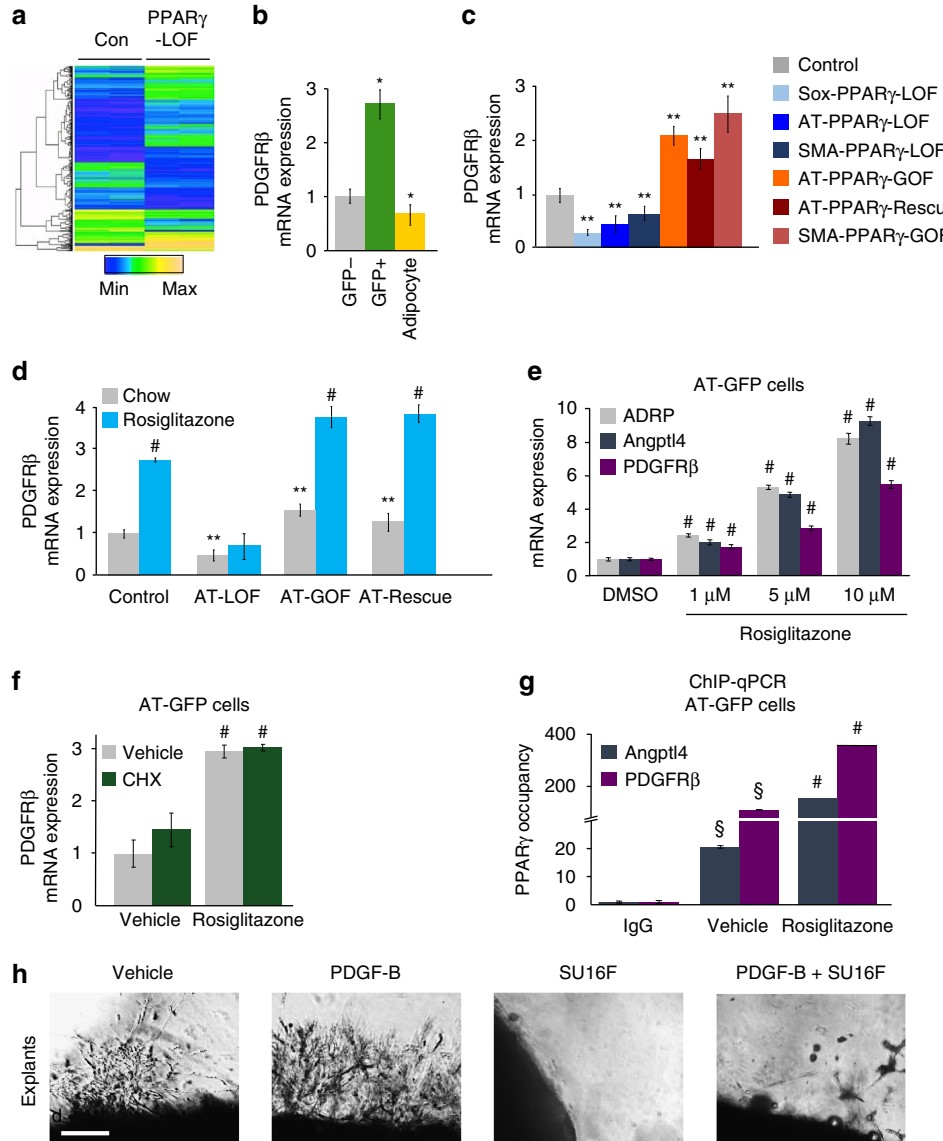

**Figure 3 | PPARγ transcriptionally controls WAT niche expansion and APC niche interaction.** (**a**) Microarray heat map of AT-GFP + (adipose progenitor) cells isolated from 2-month-old AT-control and AT-PPARγ-LOF mice (*n* = 6 mice per group in triplicate). (**b**) GFP − and GFP + cells were FACS isolated from AT-GFP control mice and adipocytes were isolated by floatation. PDGFRβ mRNA expression was measured. (**c**) GFP + cells were FACS isolated from AT-Control, Sox-PPARγ-LOF, AT-PPARγ-LOF, SMA-PPARγ-LOF and AT-PPARγ-GOF and AT-PPARγ-Rescue or SV cells were isolated from SMA-PPARγ-GOF and PDGFRβ mRNA expression was assessed. (**d**) Control, Sox-PPARγ-LOF, AT-PPARγ-LOF, AT-PPARγ-GOF and AT-PPARγ-Rescue (*n* = 10 per group) were administered normal chow or rosiglitazone (0.0075% diet) for 7 days. Subsequently, GFP + cells were FACS isolated and PDGFRβ mRNA expression was measured. (**e**) GFP + cells were FACS isolated from AT-GFP control mice (*n* = 8) and treated with the denoted concentrations of rosiglitazone for 4 h. mRNA expression of denoted genes were measured. (**f**) GFP + cells isolated from AT-control mice (*n* = 8) were pretreated with cyclohexamide (10 μg ml$^{-1}$) for 15 min and then treated with 1 μM rosiglitazone for 4 h and PDGFRβ mRNA expression was measured. (**g**) GFP + cells were FACS isolated from AT-control mice and cultured. Cells were then treated with vehicle (dimethylsulfoxide (DMSO)) or 1 μM rosiglitazone for 4 h and then ChIP-qPCR analysis was performed to assess PPARγ occupancy. (**h**) Subcutaneous IGW explants were excised from AT-control mice and encased in Matrigel. Explants were treated with vehicle (5% DMSO), PDGF-B (10 ng ml$^{-1}$), SU16F (5 μM) or both for 5 days and vascular sprouting was assessed. Scale bar 100 μm. Data are expressed as means ± s.e.m. *$P < 0.05$ unpaired *t*-test, two tailed: GFP + and adipocytes compared to GFP − SV cells. **$P < 0.05$ unpaired *t*-test, two tailed: mutants compared to AT-control mice. #$P < 0.01$ unpaired *t*-test, two tailed: rosiglitazone treated compared to chow treated. §$P < 0.01$ unpaired *t*-test, two tailed: vehicle compared to IgG control.

***Pdgfrβ* is a direct transcriptional target of PPARγ.** To garner molecular insight into how PPARγ may control adipose niche expansion and APC niche occupancy, we performed transcriptional profiling on FACS-isolated AT-GFP + APCs from control and AT-PPARγ-LOF mice (Fig. 3a). PPARγ deficiency altered the gene expression of cellular adhesion molecules and chemotaxis signalling pathways. Several of the identified pathways such as *Pdgfrβ* coordinate mural cell-to-endothelial interactions[30]. We

focussed on PDGFRβ for several reasons: PDGFRβ is expressed in APCs[11,13], PDGFRβ-Cre strains mark the adipose lineage[11], PDGFRβ signalling inhibits adipocyte differentiation[31], and PDGFRβ is a key mediator of mural-to-endothelial cell interaction and vascular expansion[30]. We hypothesized that PDGFRβ may function downstream of PPARγ to mediate APC niche occupancy and adipose niche formation. We first measured *Pdgfrβ* expression in AT-GFP + APCs, GFP − SV cells and

adipocytes. We found *Pdgfrβ* expression was enriched AT-GFP + APCs (Fig. 3b). In contrast, *Pdgf-b*, a PDGFRβ ligand[32], was highly expressed in the GFP − SV compartment compared to GFP + APCs and adipocytes (Supplementary Fig. 5a). We next examined *Pdgfrβ* expression in FACS-isolated GFP + APCs from all of our various PPARγ-expressing mouse models. We found that receptor levels were low in all PPARγ-LOF progenitors (validating the microarray) and high in all PPARγ-GOF progenitors (Fig. 3c). *PDGF-B* gene levels were unaltered in response to changes in *Pparγ* expression (Supplementary Fig. 5b).

The correlative relationship between PPARγ and PDGFRβ expression suggested that PDGFRβ might be a target gene of PPARγ. To assess this, we treated control, PPARγ-LOF and PPARγ-GOF mice with rosiglitazone. We found that APC *Pdgfrβ* mRNA expression was upregulated in a PPARγ- and ligand-dependent manner (Fig. 3d). However, *Pdgfrβ* expression in other tissues such as kidney and muscle was unchanged in response to rosiglitazone (Supplementary Fig. 5c,d). We next FACS-isolated AT-GFP + APCs and cultured them in several different concentrations (1, 5 and 10 μM) of rosiglitazone. We found that *Pdgfrβ* expression, as well as two known direct PPARγ target genes, *Adrp*[33] and *Angptl4* (ref. 34), were upregulated in a dose-dependent manner (Fig. 3e). *Pdgf-b* gene expression was unchanged in response to rosiglitazone treatment (Supplementary Fig. 5e). We next blocked PPARγ transcriptional activation by administering the PPARγ antagonist, GW9662 (ref. 28). GW9662 blocked the ability of rosiglitazone to upregulate the expression of *Pdgfrβ* or *Angptl4* (Supplementary Fig. 5f,g). However, rosiglitazone treatment had no effect on *Pdgfrβ* expression in isolated adipocytes (Supplementary Fig. 5h). To discern whether PDGFRβ might be a direct transcriptional target of PPARγ in AT-GFP + APCs, we pretreated APCs with cycloheximide and then treated cells with rosiglitazone. Cycloheximide treatment did not alter the rosiglitazone-induced upregulation of *Pdgfrβ* expression, suggesting that *Pdgfrβ* is a direct PPARγ target gene (Fig. 3f and Supplementary Fig. 5i). Using the nubiscan bioinformatics package[35], we identified a putative PPARγ response element (PPRE) within the first 100 bp of the *Pdgfrβ* transcriptional start site (Supplementary Fig. 5j). Chromatin immunoprecipitation (ChIP)-qPCR studies on FACS-isolated AT-GFP + APCs demonstrated that PPARγ and the retinoid x receptor RXR occupied the PPRE of both *Pdgfrβ* and *Angptl4* (ref. 36), suggesting a functional heterodimer transcriptional complex between PPARγ and RXR (Fig. 3g and Supplementary Fig. 5k). Collectively, these data suggest that PDGFRβ is direct target of PPARγ transcriptional activation within APCs.

**PDGFRβ mediates PPARγ-induced niche interaction**. To explore whether the PDGFRβ signalling cascade might regulate adipose niche formation, we treated FACS-sorted AT-GFP + APCs with vehicle or PDGF-B for 2 days. PDGF-B increased the expression of vasculogenic regulatory genes as well as known PDGFRβ target genes (Supplementary Fig. 6a). However, PDGF-B did not stimulate vasculogenic genes in cells deficient in PPARγ, which have reduced PDGFRβ (Supplementary Fig. 6a). These data led us to test the possibility that PDGFRβ signalling might regulate APC niche interaction. We isolated SVPs (the niche) from AT-GFP mice and treated them with vehicle, PDGF-B or SU16F, a high-affinity PDGFRβ inhibitor[37], for 12 h. In this orthotopic assay, PDGF-B increased GFP niche occupancy, whereas SU16F appeared to disrupt APC niche interactions (Supplementary Fig. 6b,c).

We next tested whether PDGFRβ signalling stimulated nichegenic action. To test this, we treated AT-GFP + adipose explants with vehicle, PDGF-B, SU16F or both and monitored vascular sprouting. PDGF-B stimulated vascular sprouting,

whereas SU16F blocked vascular sprouting (Fig. 3h and Supplementary Fig. 6d,e). Further, SU16F also reduced APC sprout occupancy (Fig. 3h and Supplementary Fig. 6f). Moreover, co-treating explants with both PDGF-B and SU16F reduced vascular sprouting highlighting the specificity of SU16F on PDGFRβ signalling (Fig. 3h and Supplementary Fig. 6d–f). We also treated AT- and SMA-PPARγ-deficient adipose explants with PDGF-B and found that PDGF-B was unable to induce sprouting in PPARγ-deficient adipose tissue explants (Supplementary Fig. 6g–i).

We next examined whether PDGFRβ signalling might regulate APC migratory potential. FACS-isolated AT-GFP-APCs were treated with vehicle, PDGF-B, SU16F or both for 48 h and then monitored APC migration. PDGF-B stimulated migration of AT-GFP-APCs, whereas SU16F inhibited migratory potential (Supplementary Fig. 6j,k). However, PDGF-B was unable to stimulate migration of PPARγ-deficient APCs presumably due to low levels of *Pdgfrβ* expression (Supplementary Fig. 6l–n). To test this, we reconstituted the full-length cDNA of *Pdgfrβ* in PPARγ-deficient cells and monitored migration (Supplementary Fig. 6l). Overexpressing PDGFRβ in PPARγ-LOF cells appeared to increase migration and the addition of PDGF-B enhanced this effect (Supplementary Fig. 6m,n). Taken together, these data are consistent with a model in which PPARγ transcriptionally regulates PDGFRβ, thus regulating niche expansion, APC cell migration, niche interaction and niche occupancy.

**PDGFRβ regulates niche formation and maintenance**. To further delineate whether PDGFRβ directly alters APC niche interaction, mimicking PPARγ deficiency, we integrated a *Pdgfrβ*<sup>fl/fl</sup> deletion allele (PDGFRβ-LOF) or a *Pdgfrβ* constitutively active allele (PDGFRβ-KI<sup>fl</sup> (ref. 31); PDGFRβ-CA) into our AdipoTrak system (Fig. 4a). Both PDGFRβ- LOF and CA mice had normal adipose depot formation at P10 (Supplementary Fig. 7a–f). However, by P60 both adult mutant mice had a 50% reduction in fat content and all adipose depots were smaller compared to controls (Supplementary Fig. 7g–i). Histological examination of adipose depots from both mutant mice revealed disrupted adipose tissue architecture, small adipocytes, enlarged blood vessels and extensive interstitial staining (Fig. 4b); however, both mutant mice maintained their glucose sensitivity and had normal liver histology (Supplementary Fig. 7j,k). We next examined APC locality and found that PDGFRβ-deficient APCs resided off the vasculature, resembling the PPARγ-LOF mice (Fig. 4c). In contrast, PDGFRβ-CA APCs appeared locked at the vascular interface (Fig. 4c). SVPs confirmed these findings: very few PDGFRβ-LOF APCs were associated with the vessel fragment, whereas the PDGFRβ-CA had many (Fig. 4d and Supplementary Fig. 7l). We also assessed niche formation and found that AT-PDGFRβ-LOF explants had decreased vascular sprouting, sprout length, branching and APC occupancy, whereas PDGFRβ-CA had many more sprouts and higher APC occupancy (Fig. 4e and Supplementary Fig. 7m,n). Further, the addition of PDGF-B ligand to AT-PDGFRβ-LOF adipose tissue explants did not rescue the vascular sprouting defect (Supplementary Fig. 7n). We also found that angiogenic markers (*Vegf*, *Fgf2* and *Angptl4*) were lower in the PDGFRβ-LOF but were elevated in PDGFRβ-CA adipose depots (Fig. 4h). We also found that APCs deficient in PDGFRβ did not migrate in the absence or presence of PDGF-B (Supplementary Fig. 7o). Together, these data suggest that PDGFRβ plays a crucial role in APC niche recruitment and niche formation.

**PDGFRβ regulates APC niche retention**. One caveat of the AdipoTrak-system is that it marks the entire adipose lineage

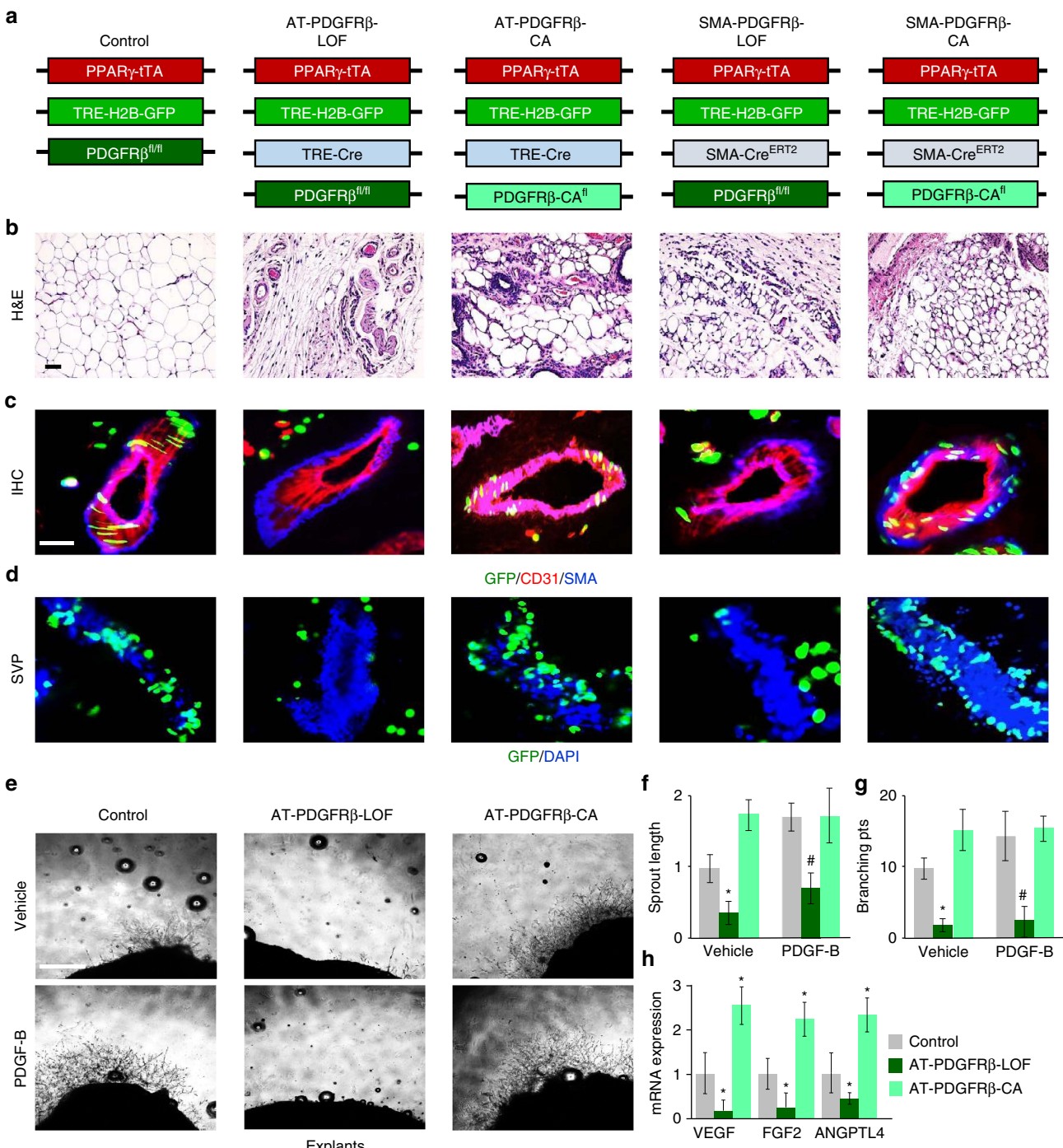

**Figure 4 | PDGFRβ regulates APC niche interaction and WAT niche expansion.** (**a**) Illustration of genetic alleles used to generate AT- and *SMA-Cre*^*ERT2*-*PDGFRβ* loss (AT-PDGFRβ-LOF or SMA-PDGFRβ-LOF) and constitutively active PDGFRβ (AT-PDGFRβ-CA or SMA-PDGFRβ-CA) mice. Experiments were performed three times on 10 mice per group. (**b**) Representative H&E images of IGW adipose depots from mice described in **a**. (**c**) Representative histological sections from mice described in **a** stained for GFP, CD31 and SMA. (**d**) Representative images of SVPs isolated from mice described in **a**. (**e**) Representative images of vascular sprouts of subcutaneous IGW explants from the mice described in **a**. (**f,g**) Sprout length and branching points were quantified from explants described in **e**. (**h**) mRNA expression of vasculogenic genes from mice described in **a**. Data are expressed as means ± s.e.m. Scale bars 100 μm. *P < 0.01 unpaired *t*-test, two tailed: mutant compared to control levels. #P < 0.001 unpaired *t*-test, two tailed: mutant PDGF-B treated compared to control PDGF-B treated.

(stem to adipocyte). Thus PDGFRβ could alter APC niche biology via other mechanisms. To interrogate the adult APC compartment and in intact adipose depots, we combined our AT-GFP tracking system with *SMA-Cre*^*ERT2*; *R26R*^*RFP*^ and *Pdgfrβ*^fl/fl^ (LOF) or *Pdgfrβ*^+/Kfl^ (PDGFRβ-CA) mice (Fig. 4a). We administered TM and evaluated these mice 4 weeks later

(Supplementary Fig. 8a). PDGFRβ deletion or activation in the SMA-APC compartment prevented fat mass accumulation, led to smaller adipose depots with enlarged blood vessels but normal glucose sensitivity (Fig. 4b and Supplementary Fig. 8b–h). Histological examination of other tissues revealed normal histology without disruption (Supplementary Fig. 8i). SMA-RFP fate

mapping studies, a surrogate of new adipocyte formation, showed that neither PDGFRβ-LOF nor PDGFRβ-CA generated new white adipocytes after the 4-week chase (Supplementary Fig. 8g). Progenitor niche interaction tests revealed that SMA-PDGFRβ-LOF APCs were no longer retained at the blood vessel (Fig. 4c). In contrast, SMA-PDGFRβ-CA GFP + APCs were locked at the vascular interface (Fig. 4c). SVPs showed similar results: reduced number of GFP + APCs/SVP in the LOF model and more AT-GFP-APCs/SVP in the CA model (Fig. 4d and Supplementary Fig. 8j). Moreover, SMA-PDGFRβ-LOF WAT explants had impaired vascular sprouting and PDGF-B addition had no effect on sprouting (Fig. 4e–h and Supplementary Fig. 8k,l). In contrast, SMA-PDGFRβ-CA mice had robust sprouting but was unaffected by PDGF-B likely due to constitutive active signalling. PDGFRβ loss also appeared to inhibit APC dynamics, such as migration rate (Supplementary Fig. 8m). Taken together, it appears that PDGFRβ is required for APC niche interaction and retention and cellular migration of APCs and this interaction mediates adipogenesis.

**PDGFRβ restores niche function under PPARγ deficiency.** The above studies suggested that PDGFRβ signalling could rescue the dysfunctional progenitor cell dynamics of AT-PPARγ-LOF APCs such as migration. Therefore, we combined the $Pdgfrβ^{+/Kfl}$ constitutively active mouse model (PDGFRβ-CA) with our AT-PPARγ-LOF mouse model (Fig. 5a). Remarkably, expressing the PDGFRβ-CA allele appeared to rescue the PPARγ-LOF APC vascular niche mis-localization (Fig. 5b,c,e). Remarkably, PDGFRβ kinase activity re-established niche expansion action in AT-PPARγ-LOF compared to AT-PPARγ-LOF alone (Fig. 5d,f). Consistently, PDGFRβ activation restored WAT vasculogenic gene expression in the AT-PPARγ-LOF mouse model (Fig. 5g).

**In vivo pharmacologically targeting PDGFRβ signalling.** Our in vivo and in vitro studies support the notion that PDGFRβ signalling mediates APC niche interaction downstream of PPARγ. We next examined whether treating mice with imatinib (Gleevec), a Food and Drug Administration-approved PDGFRβ inhibitor that also targets other tyrosine kinase cascades, could alter APC cell niche locality and adiposity[38,39]. We treated 1-month-old AT-GFP control mice with vehicle (5% dimethylsulfoxide) or imatinib (50 μg per mouse) for 4 weeks (Supplementary Fig. 9a). Body weight progressively increased and food intake was unaltered between vehicle and imatinib-treated mice (Fig. 6a,b). Strikingly, imatinib treatment improved glucose sensitivity, an affect also observed in chronic myeloid leukaemia patients prescribed with imatinib (Supplementary Fig. 9b,c)[38]. Notably, imatinib administration blunted subcutaneous and visceral fat expansion (Fig. 6c–e). Other tissues appeared to be unaffected by imatinib treatment except skeletal muscle, which showed slightly increased weight (Supplementary Fig. 9d). WAT histological examination revealed smaller adipocytes but normal adipose tissue architecture; however, in some imatinib-treated sections blood vessels appeared irregular (Fig. 6f). Flow cytometry and qPCR indicated that imatinib did not alter the number or gene expression of endothelial or mural markers (Supplementary Fig. 9e,f). However, imatinib treatment displaced AT-GFP + APCs from their vascular niche (Fig. 6g). In agreement, the distance of AT-GFP + APCs from CD31 + vessels were increased in response to imatinib treatment, indicating disruption of APC niche residency (Fig. 6h). Consistently, SVPs showed similar results (Fig. 6i,j). Of note, imatinib treatment did not alter AT-GFP + APC number or the apoptotic marker,

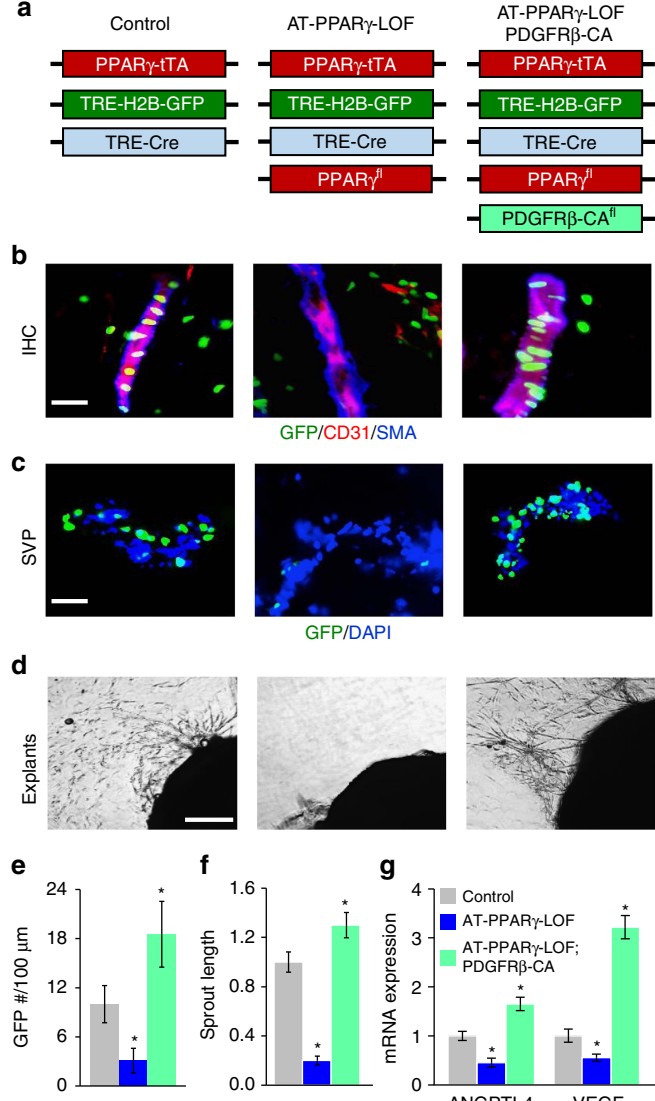

**Figure 5 | PDGFRβ restores APC niche interaction and expansion in AT-PPARγ-LOF mice.** (**a**) Illustration of genetic alleles used to generate AT-control, AT-PPARγ-LOF and AT-PPARγ-LOF-PDGFRβ-CA mice. At 2 months of age, mice were analysed. Experiments were performed three times on 10 mice per group. (**b**) Representative images of subcutaneous IGW depots stained for CD31, SMA and APC-GFP from mice described in **a**. (**c**) Representative image of SVPs isolated from mice described in **a** and visualized for GFP locality. (**d**) Representative images of vascular sprouts from subcutaneous IGW WAT explants excised from mice described in **a**. (**e**) Quantification of APC-GFP number per 100-micron SVP. (**f**) Quantification of sprout length of explants described in **d**. (**g**) Quantitative RT–PCR analysis of vasculogenic gene expression from mice described in **a**. Data are expressed as means ± s.e.m. Scale bars 100 μm. *$P < 0.05$ unpaired t-test, two tailed: mutant compared to control levels.

cleaved caspase 3 (Supplementary Fig. 9g,h). Niche explant assays showed that imatinib blunted vascular sprouting and branching and reduced progenitor occupancy (Fig. 6k–m). In conjunction, angiogenic markers were also downregulated in response to imatinib treatment (Supplementary Fig. 9i). Based on these results, imatinib treatment disrupts the ability of APC cells to interact with their vascular niche and blocked white adipocyte differentiation.

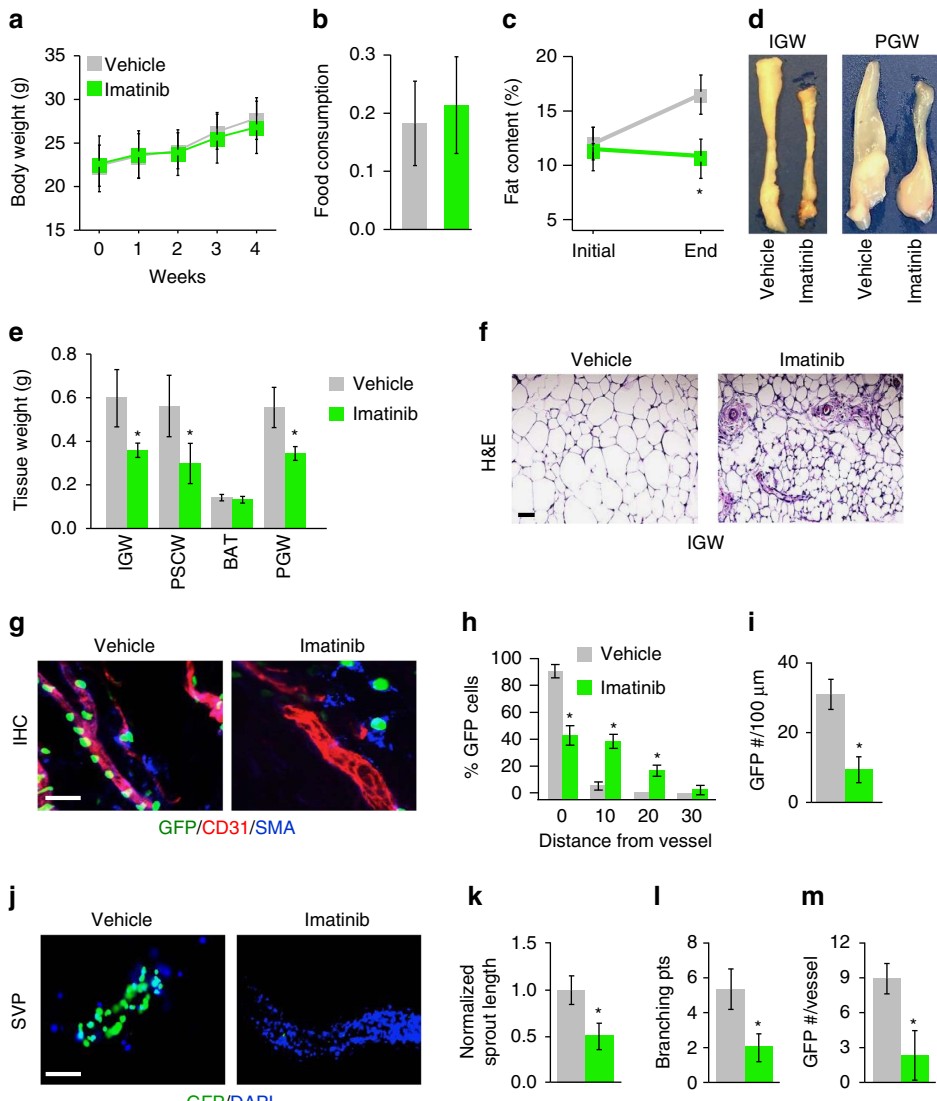

**Figure 6 | Pharmacologically blocking PDGFRβ disrupts APC niche interaction.** (**a,b**) One-month-old AT-GFP male mice were administered vehicle (5% dimethylsulfoxide (DMSO)) or imatinib (50 μg per mouse) by IP four times a week for 4 weeks. Experiments were performed three times on 8 mice per group. Body weight (**a**) and food intake (**b**) were measured. (**c**) Fat content of mice described in **a** before and end of treatment regime. (**d,e**) Representative adipose tissue images (**d**) and weights (**e**) from mice described in **a**. (**f**) Representative images of H&E staining from subcutaneous IGW depots from mice described in **a**. Scale bar 100 μm. (**g**) Sections from subcutaneous IGW depots from mice described in **a** were stained with CD31 and SMA and visualized for AT-GFP. Scale bar = 100 μm. (**h**) Quantification of distance of AT-GFP + cells away from CD31/SMA + blood vessels from sections described in **g**. (**i,j**) SVPs were isolated from mice described in **a** and cultured. GFP locality was visualized 12 h later (**j**) and AT-GFP number was quantified (**i**). Scale bar = 100 μm. (**k–m**) Subcutaneous IGW depots from mice described in **a** were excised and encased in Matrigel. Explants were continually treated with vehicle or imatinib *ex vivo* for 5 days. Vascular sprouts were then quantified for sprout length (**k**), branching (**l**) and progenitor occupancy (**m**). *P < 0.01 Imatinib treated compared to vehicle (DMSO) treated. Data are expressed as means ± s.e.m. Scale bars 100 μm.

**VEGF mediates PPARγ-induced niche expansion.** We next probed a possible mechanism by which APC *Pparγ* expression could mediate endothelial proliferation and vasculogenic action within WAT. Throughout our experiments, *Vegf* expression correlated with *Pparγ* expression. White adipocyte VEGF expression is known to have a role in WAT vascular expansion[40–42]; however, our studies suggest that APC VEGF expression may mediate WAT vascular niche expansion. Expression analysis demonstrated that *Vegf* mRNA was enriched in AT-GFP-APCs compared to adipocytes and AT-GFP-negative SV cells (Fig. 7a). Next, FACS-isolated AT-GFP-APCs were treated with recombinant VEGF; overall, VEGF did not appear to alter APC proliferation or migration (Fig. 7b,c). Yet, treating total SV cells with VEGF increased the expression of endothelial markers and enhanced the

vascular assembly networks (Fig. 7d–f). Further, AT-GFP-APCs appeared to co-localize with these newly formed VEGF-stimulated vessels (Fig. 7g). WAT explants from *Tie2-Cre; Rosa26R^RFP* (endothelial cell marker[43]) mice confirmed these findings: robust vascular sprouting in the presence of recombinant VEGF (Supplementary Fig. 10a).

Previous studies have proposed that VEGF is a PPARγ direct transcriptional target gene[29,44]. To evaluate this notion, we tested whether *Vegf* expression correlated in our various PPARγ-LOF and -GOF models. *Vegf* expression was associated with PPARγ expression and in a ligand-dependent manner (Fig. 7h and Supplementary Fig. 10b). We then tested whether *Vegf* was a direct PPARγ target gene in AT-GFP-APCs by transcriptional activation, cycloheximide and ChIP-qPCR studies (Fig. 7i–k).

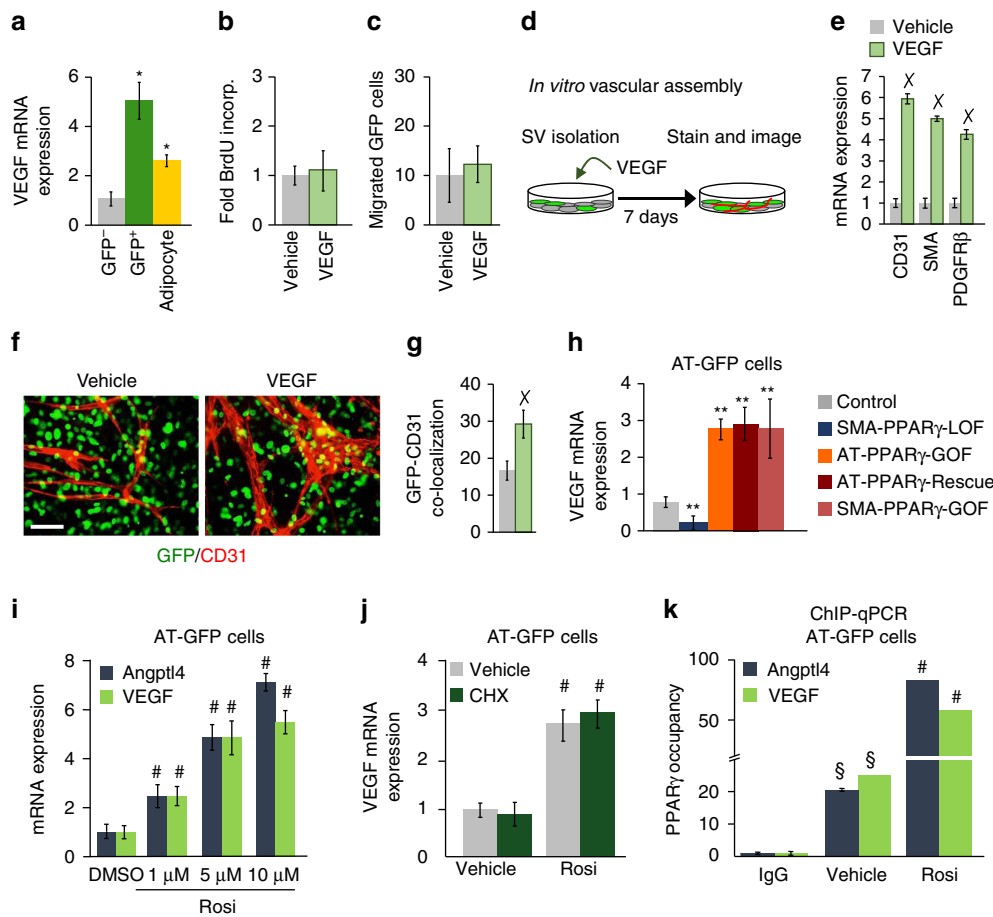

**Figure 7 | VEGF is a transcriptional target of PPARγ in APCs.** (**a**) GFP − and GFP + cells were FACS isolated from AT-GFP control mice and adipocytes were isolated by floatation ($n = 6$). VEGF mRNA expression was measured. (**b**) AT-GFP + cells were FACS isolated from AT-GFP control mice ($n = 6$) and treated with vehicle or VEGF for 8 h and then treated with BrdU for 12 h. BrdU incorporation was monitored. (**c**) AT-GFP + cells were FACS isolated from AT-GFP control mice ($n = 6$) and treated with vehicle or VEGF for 8 h and then plated in transwell migration chambers. Migration was monitored 12 h later and quantified. (**d–f**) Total SV cells were isolated from AT-GFP control mice ($n = 6$). Cells were treated with vehicle or VEGF for 12 h and then mRNA was harvested or cells were imaged for AT-GFP and stained for CD31. (**g**) Cells and staining described in **d** were quantified for AT-GFP and CD31 co-localization. (**h**) AT-GFP + cells were FACS isolated from AT-control, SMA-PPARγ-LOF and AT-PPARγ-GOF and AT-PPARγ-Rescue or SV cells were isolated from SMA-PPARγ-GOF and VEGF mRNA expression was assessed ($n = 6$ per group). (**i**) AT-GFP + cells were FACS isolated from AT-control mice ($n = 6$) and treated with denoted concentrations of rosiglitazone for 4 h. mRNA expression of VEGF and Angptl4 were measured. (**j**) AT-GFP + cells isolated from AT-control mice ($n = 6$) and were pretreated with cyclohexamide ($10 \mu g \, ml^{-1}$) for 15 min and then treated with $1 \mu M$ rosiglitazone for 4 h and PDGFRβ mRNA expression was measured. (**k**) AT-GFP + cells were FACS isolated from AT-control mice ($n = 6$). Cells were then treated with vehicle (dimethylsulfoxide (DMSO)) or $1 \mu M$ rosiglitazone for 4 h and then ChIP-qPCR analysis was performed to assess PPARγ occupancy. Data are expressed as means ± s.e.m. Scale bar 100 μm. *$P < 0.05$ unpaired $t$-test, two tailed: GFP + and adipocytes compared to GFP − SV cells. χ $P < 0.01$ unpaired $t$-test, two tailed: VEGF treated compared to vehicle treated cells. **$P < 0.02$ unpaired $t$-test, two tailed: mutant mice compared to control mice. §$P < 0.001$ unpaired $t$-test, two tailed: vehicle compared to IgG control. #$P < 0.001$ rosiglitazone treated compared to vehicle (DMSO) treated.

We found that PPARγ directly transcriptionally regulates *Vegf* expression in AT-GFP-APCs.

We next tested whether enhanced VEGF expression mediates APC niche interaction and niche expansion. Towards this end, we incorporated a *Tre-Vegf* allele[40] with our AT (AdipoTrak-tTA) adipose lineage system termed AT-VEGF (Supplementary Fig. 11a). By 3 months of age, AT-VEGF adipose depots were 70% smaller and had reduced mRNA expression of adipocyte markers compared to AT-controls (Fig. 8a and Supplementary Fig. 11b,c). However, these lipodystrophic mice maintained normal-fed sera glucose levels (Supplementary Fig. 11d). Histological assessments of AT-VEGF adipose depots revealed many blood vessels with very few adipocytes (Fig. 8a,b and Supplementary Fig. 11e–g). Immunostaining demonstrated that AT-GFP-APCs were strongly associated with the vasculature (Fig. 8c). AT-VEGF SVPs showed similar results: many AT-GFP-

APCs interacting with blood vessel fragments (Fig. 8d and Supplementary Fig. 11h). Vascular and nichegenic assays showed that AT-VEGF WAT explants had higher density of vascular sprouts, increased sprout length and many more branch points (Fig. 8e and Supplementary Fig. 11i,j). Molecular analysis revealed enhanced vasculogenic and nichegenic markers (Fig. 8f,g). Flow cytometric studies also indicated that CD31 + and SMA + cell numbers were increased in AT-VEGF mice (Supplementary Fig. 11k,l). Consistently, AT-GFP-APC number was also increased (Supplementary Fig. 11m). We next examined endothelial cell proliferation by bromodeoxyuridine (BrdU) pulse labelling of AT-control and AT-VEGF mice for 12 h. Endothelial cells that were FACS isolated from AT-VEGF showed an increase in BrdU incorporation compared to controls (Supplementary Fig. 11n). To test whether this enhanced nichegenic potential was a progenitor-driven effect, total SV cells were isolated from

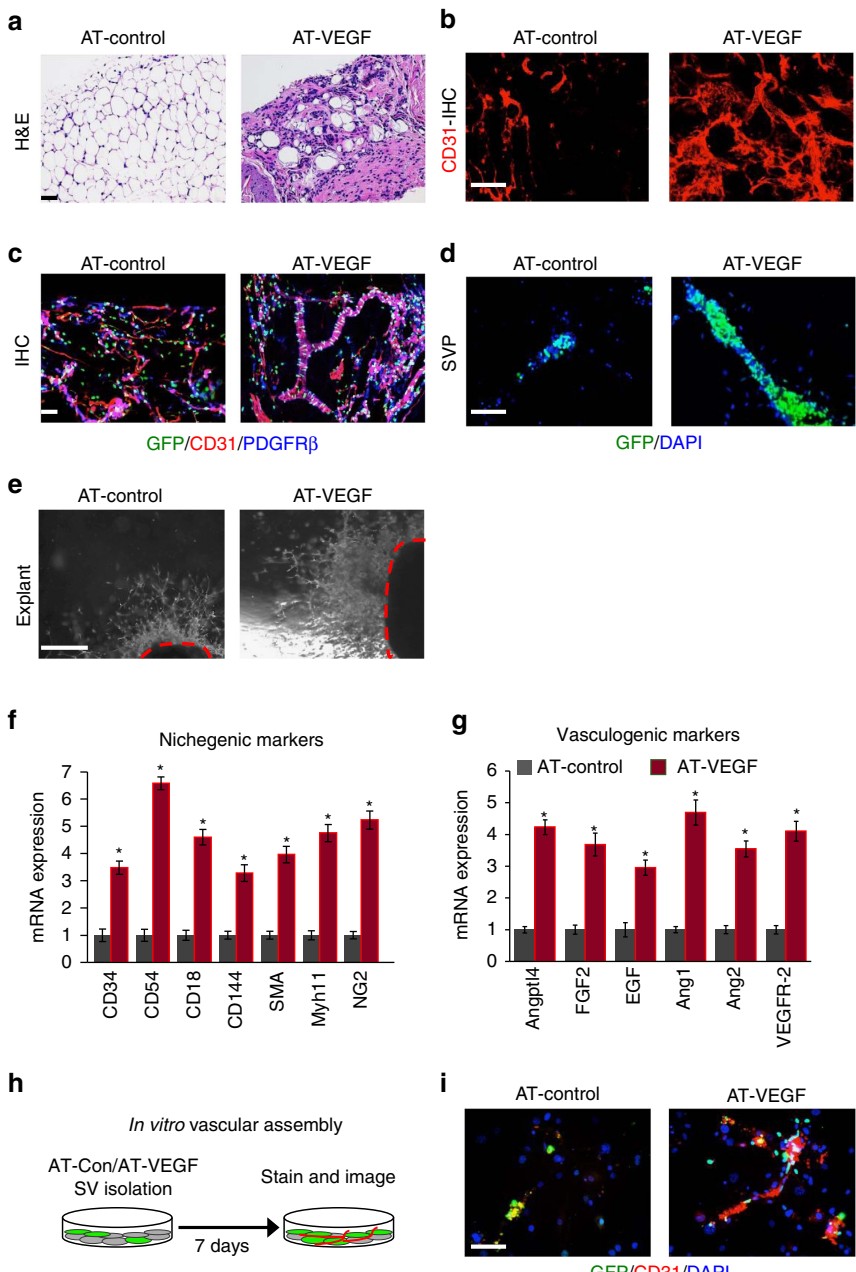

**Figure 8 | VEGF stimulates niche expansion but blocks fat formation.** (**a**) Representative images of H&E staining of AT-control (*PPARγ^tTA; TRE-H2B-GFP*) and AT-VEGF (*PPARγ^tTA; TRE-VEGF; TRE-H2B-GFP*) at 3 months of age. (**b**) Representative images of CD31 immunostaining of IGW adipose depots from AT-control and AT-VEGF mice. (**c**) AT-control and AT-VEGF IGW adipose depots were sectioned and stained for GFP, CD31 and SMA. (**d**) Representative images of SVPs isolated from 3-month old AT-control and AT-VEGF mice and analysed for GFP locality. (**e**) Representative images of vascular sprouts from subcutaneous IGW adipose explants from 3-month old AT-control and AT-VEGF mice. (**f,g**) Quantitative RT–PCR analysis of niche and vasculogenic genes from 2-month old AT-control and AT-VEGF mice. (**h,i**) Total SV cells were isolated from AT-control and AT-VEGF mice ($n = 6$). Cells were cultured for 7 days and then examined for vascular assembly by examining CD31 staining and GFP + cell co-localization. Scale bars 100 μm. Data are expressed as means ± s.e.m. Experiments were performed three times on 6–10 mice per group. Scale bars 100 μm. *$P < 0.05$ unpaired *t*-test, two tailed: mutant compared to control levels.

AT-control and AT-VEGF mice and then cultures were monitored for vascular assembly. We found that AT-VEGF cultures contained more CD31 + cells and assembled a vascular network within the culture setting (Fig. 8h,i).

To test whether VEGF was required to sustain the WAT vascular phenotype, we administered Dox, to suppress VEGF expression, to 2-month-old AT-control and AT-VEGF male mice for 30 days (Supplementary Fig. 12a). We found that body weight remained similar between controls and AT-VEGF Dox-suppressed mice. However, fat content increased by twofold when VEGF was suppressed compared to unsuppressed VEGF-overexpressing mutants (Supplementary Fig. 12b,c). This was also confirmed by histology and qPCR studies (Supplementary Fig. 12d,e). In addition, the vascular phenotype observed in AT-VEGF mice was completely restored to normal after 1 month of Dox suppression (Supplementary Fig. 12f,g). Collectively, these studies

suggest that continued VEGF expression is required to tether APCs to the niche, maintain the progenitor state and expand the niche.

**APC VEGF expression stimulates WAT niche expansion.** We next attempted to temporally control VEGF expression within the adult APC compartment by combining the Dox-inducible *SMA-rtTA* APC-mural driver with the *Tre-Vegf* allele (SMA-rtTA-VEGF) to test whether VEGF, within the APC compartment, is sufficient to drive WAT vascular expansion and APC niche tethering. To induce VEGF expression in adult APCs, we administered Dox to 2-month-old SMA-rtTA-control and SMA-rtTA-VEGF mice for 14 days (Fig. 9a). Histological sections revealed that SMA-driven VEGF expression stimulated acute WAT blood vessel expansion (Fig. 9b). SMA-driven VEGF expression also increased gene expression of the endothelial cell marker, *Cd31*, in WATs and in other SMA+ organs, such as the skeletal muscle (Fig. 9c). Flow cytometric studies indicated that CD31 and SMA cell numbers were also increased in SMA-rtTA-VEGF WAT depots compared to controls (Fig. 9d,e). BrdU pulse studies demonstrated that more CD31+ cells were BrdU+ within SMA-rtTA-VEGF WAT compared to CD31+ cells in SMA-control WAT (Fig. 9f). These findings correlated with the upregulation of nichegenic and vasculogenic gene expression in SMA-rtTA-VEGF mice (Fig. 9g–i). Taken together, our data

suggest that overexpressing VEGF from the adult APC mural compartment is sufficient to stimulate endothelial cell proliferation thereby expanding the WAT APC vascular niche.

## Discussion

In many organ systems, stem cells reside in a specialized microenvironment termed the niche[45–49]. Our studies raise the possibility that APCs, labelled by the AdipoTrak system, play key roles in the genesis, elaboration and maintenance of their own WAT niche. These functions of APC on niche biology appear to rely, at least in part, on PPARγ to PDGFRβ and PPARγ to VEGF molecular networks. PPARγ is a nuclear hormone receptor activated by various fatty acids and by the thiazolidinedione class of insulin sensitizers that have been widely prescribed for diabetes[50,51]. PPARγ is a master regulator of adipocyte differentiation[51]. However, our data indicate that PPARγ has niche and APC roles in the adipose lineage in addition to adipocyte maturation. Through GOF and LOF strategies, we found that PPARγ controls APC migration, vascular niche formation and APC occupancy of the niche. We deleted PPARγ in APCs before (*Sox2-Cre*; *AdipoTrak-Cre*) and after (*Sma-Cre^ERT2*) they entered the niche. In the former setting, the APCs did not acquire niche localization and appeared juxtaposed to rather than embedded in the blood vessel. In the adult setting

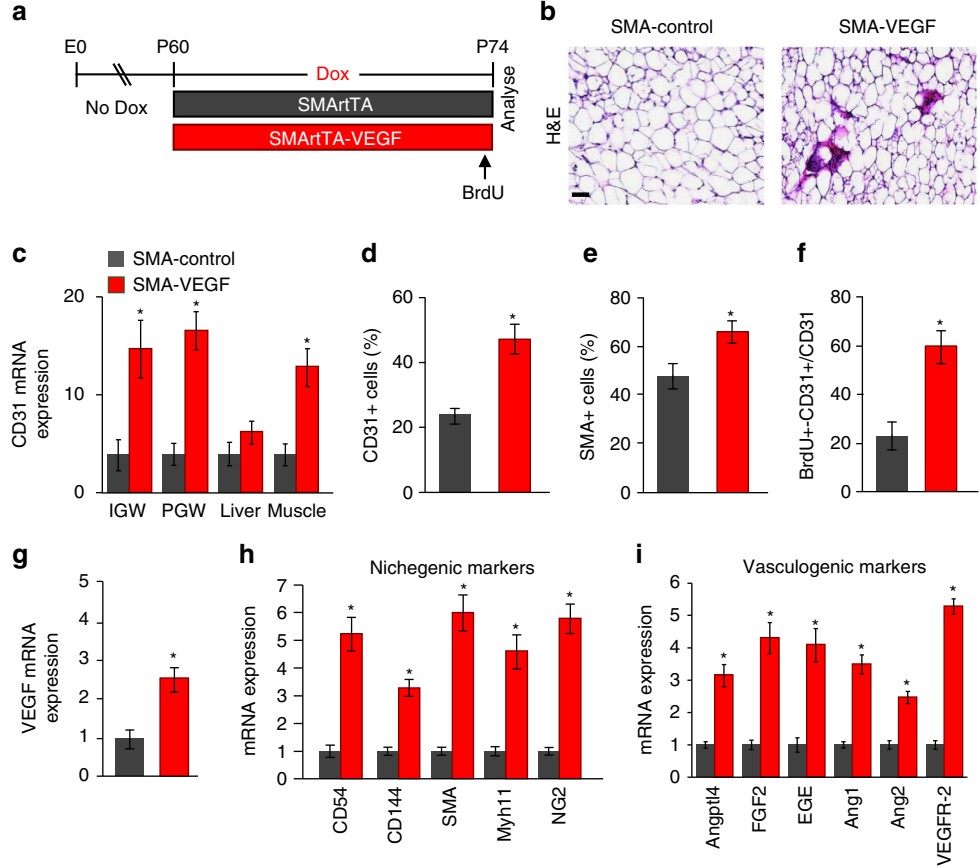

**Figure 9 | VEGF stimulates niche expansion from adult APCs.** (**a**) Diagram of experimental paradigm. At P60, *SMA-rtTA* control or *SMA-rtTA; TRE-VEGF* (SMArtTA-VEGF) mice were administered Dox for 2 weeks. BrdU was administered 24 h before final analysis. Experiments were performed three times on seven mice per group. (**b**) Representative images of H&E staining of SMArtTA-control and SMArtTA-VEGF mice described in **a**. (**c**) Quantitative RT–PCR analysis of CD31 mRNA expression from denoted tissues. (**d–f**) Total SV cells were isolated from subcutaneous IGW depots mice described in **a**. Cells were stained and examined for CD31 (**d**), SMA (**e**) or BrdU and CD31-positive cells by flow cytometry. (**g–i**) Quantitative RT–PCR analysis of VEGF mRNA expression (**g**), nichegenic (**h**) and vasculogenic (**i**) genes from mice described in **a**. Scale bar 50 μm. Data are expressed as means ± s.e.m. *$P < 0.01$ unpaired *t*-test, two tailed: mutant compared to control levels.

(SMA), after the APCs acquire niche residency, SMA-APC deletion of PPARγ disrupts niche occupancy; that is, they 'fall out' of the niche. However, a potential limitation of these studies is the ability to discern precise differences between PPARγ's adipocyte differentiation role and its progenitor cell role. Yet, both *in vitro* and *ex vivo* models do suggest that PPARγ regulates progenitor cell properties but future studies and new genetic models could assist in interrogating PPARγ's APC function from its adipogenic role.

PDGFRβ is an important regulator of mural cell biology that creates a bridge between mural and endothelial cells, thereby stimulating vascular growth[52,53]. We found that *Pdgfrβ* is a direct PPARγ transcriptional target. Disruption of this mural–endothelial bridge decreases vascular stability, reduces endothelial junctions, promotes haemorrhage and decreases overall angiogenesis[30]. In line with these functions, we found that PDGF-B stimulates WAT vascular expansion, APC cell migration and APC cell vascular niche occupancy. However, in the PPARγ-LOF settings, where PDGFRβ expression is lower, PDGF-B's ability to stimulate APC niche interaction and vascular expansion is minimal, supporting the notion that APCs can expand their niche. Further our studies indicate that APC have a direct role in communicating with their niche. This direct line of communication allows them to control the construction, maintenance and expansion of their own niche. However, what is the direct line of communication emanating from PDGFRβ that allows APCs to regulate angiogenic action? Our transcriptional data hint that a vasculogenic programme is engaged when PPARγ-PDGFRβ network is activated. Moreover, the data suggest that APC niche interaction and niche expansion occurs simultaneously. That is without APCs WAT vascular expansion is halted and vice versa without vascular expansion APC proliferation, migration and differentiation is impeded. Lending further credence to this notion is the effects of PDGFRβ inhibitors, both *in vitro* and *in vivo*, that appear to alter WAT niche biology. When PDGFRβ inhibitors such as SU16F, a relatively restricted PDGFRβ inhibitor[37], or imatinib (Gleevec)[38,39], a Food and Drug Administration-approved tyrosine-kinase inhibitor, were administered either *in vitro* or *in vivo*, APCs did not interact with their vascular niche and did not occupy niche positions, similar to our PDGFRβ genetic studies. Further, APCs were unable to undergo adipogenic differentiation *in vivo* resulting in disrupted adipose tissue expansion. Remarkably, *in vivo* administration did not induce changes in systemic glucose metabolism, rather glucose sensitivity was improved, which is commonly observed in chronic myeloid leukaemia patients after imatinib therapy[54–57]. Of note, although imatinib does inhibit PDGFRβ signalling, it also affects other tyrosine kinase signalling cascades (for example, ABL, KIT, PDGFRα)[58]. Additionally, are the effect of glucose sensitivity due solely to WATs and APC niche mis-localization remains unanswered.

PPARγ also seems to be critical for vascular niche expansion[24,29]. PPARγ's ability to expand the vascular niche appears to rely, at least in part, on its transcriptional role to upregulate *Vegf* expression within APCs. VEGF is then secreted from APCs and stimulates endothelial cell proliferation and vascular assembly. Canonical VEGF upregulation requires hypoxic conditions[59]; however, whether this includes PPARγ or whether this is a hypoxia-independent action of neovascularization remains to be explored? Future studies examining this role of hypoxia could provide significant insight between the interplay between APCs and niche stimulation.

In summary, these studies support and expand a model in which APCs are critical mediators of their own niche and that PPARγ is a central regulator of this communication network.

Mechanistically, PPARγ transcriptionally drives the regulation of genes such as *Pdgfrβ* rendering the cells responsive to PDGF-B, which enhances the APC niche interaction and promotes niche assembly and expansion. Furthermore, PPARγ initiates and directs WAT expansion by transcriptionally regulating APC *Vegf* expression and VEGF secretion. These studies represent the potential role that APCs have in regulating their niche and that there is a bi-directional communication network between niche and progenitors.

## Methods

**Mice.** AdipoTrak (PPARγ[tTA]; TRE-H2B-GFP; TRE-Cre) mice (C57BL/6-129/SV) were previously established in our laboratory[11,22,27]. *Sox2-Cre* (stock no: 014094), *PPARγ[fl/fl]* (stock no: 004584), *PDGFRβ[fl/fl]* (stock no: 010977), *PDGFRβ-KI[fl]* (stock no: 018435), *Tie2-Cre* (stock no: 008863) and the reporter line *R26R[RFP]* (stock no: 007908) were purchased from the Jackson Laboratory. TRE-VEGF mice were generously provided by Dr Eli Keshet[40]. All mouse lines were backcrossed on a C57BL/6-129/SV mixed background for at least 10 generations. *Tre-PPARγ* mice were generated as previously described[25]. Briefly, we introduced p2Lox-PPARγ into ZX1 ES cells and selected for G418-resistant recombinant colonies, in which PPARγ was recombined downstream of a tetracycline responsive promoter (TRE). ZX1 ES cell clones were then validated in culture and three clones were used to generate three chimeric mouse lines. All three mouse strains behaved similarly and had similar induction of PPARγ expression in response to Dox. *SMA-Cre[ERT2]* was generously provided by Pierre Chambon[60]. Cre[ERT2]-mediated recombination was induced by administering TM (Sigma, item no.: T5648) (50 mg kg$^{-1}$ intraperitoneal injection) dissolved in sunflower oil (Sigma, item no.: S5007) on 2 consecutive days. Dr Beverly Rothermel generously provided *SMA-rtTA* mice[12]. To induce activation of the *rtTA*, mice were administered Dox (Cayman Chemicals: item no.: 14422) (0.25 mg l$^{-1}$) in drinking water. Dox water was protected from light and changed 3× per week. Mice were fed rosiglitazone (Cayman Chemicals: item no: 71740) at 0.0075% in normal chow (4% fat mouse diet from Harlan TEKLAD) *ad libitum*. Rosiglitazone intake was estimated to be 15 mg kg$^{-1}$ body mass per day[27]. Imatinib treatment: mice were administered imatinib (Cayman Chemicals: item no.: 13139) 50 µg per mouse by intraperitoneal injection or vehicle (5% dimethylsulfoxide in water). Mice were administered vehicle or imatinib four times a week for 4 weeks. Body weight was monitored weekly using a Mettler Toledo scale. Food intake was measured by weighing the initial food weight and weighing the remaining food after 24 h (ref. 10). Experiments were performed on male and female mice as indicated. All animal procedures were ethically approved by the UT Southwestern Medical Center institutional animal care and use committee.

**Histology.** Adipose tissues were fixed in formalin overnight, dehydrated, embedded in paraffin and sectioned with a microtome at 5–8 µm thicknesses. H&E staining was carried out on paraffin sections using standard methods. Immunostaining was performed in either paraffin sections or cryostat sections (5–8 µm) of tissues freshly embedded in Optimal Cutting Temperature as previously described[11]. Briefly, samples were preincubated with permeabilization buffer (0.3% Triton X-100 in PBS) for 30 min at room temperature and then incubated sequentially with primary antibody (4 °C, overnight) and secondary antibody (2 h at room temperature), all in blocking buffer (5% normal donkey serum in 1× PBS). Antibodies used for immunostaining are: chicken-anti-GFP (1:1,000, Abcam: ab13970), rat-anti-CD31 (1:200, BD Biosciences: item no.: 550274), rabbit-anti-α-SMA (1:500, Sigma: item no.: SAB5500002), and rabbit-anti-PDGFRβ (1:500, eBioscience: item no.: 14-1402-82). Secondary antibodies including Alexa488 donkey anti-chicken (item no.: 703-545-155), cy5 donkey anti-rabbit (item no.: 711-175-152) and cy3 donkey anti-rat (item no.: 712-175-150) were from Jackson ImmunoResearch. All secondary antibodies were used at a 1:500 dilution. Lipid was stained with LipidTOX Red (1:200, Life Technologies: item no.: H34476) on cryosections or tissue whole mounts. Immunostained images were collected on a Zeiss LSM500 confocal microscope, a Lecia DMi8 inverted microscope or a Lecia DM6B upright microscope. Paraffin-embedded tissues were sectioned with a Microm HM 325 microtome. Cryostat sectioning was performed with a Microm HM505 E cryostat. Sections were quantitated for APC cell locality using the Image J software from NIH.

**SV and SVP fractionation.** To fractionate SV cells, adipocytes and SVP, we pooled subcutaneous (inguinal, interscapular) or visceral (gonadal and retroperitoneal) WATs. After 2 h of slow shaking at 37 °C, the suspension was spun at 800 g for 10 min; the resultant floating layer was the adipocyte layer and the pellet was a crude SV fraction. The floated adipocyte layer was collected for RNA extraction. The SV pellet was then resuspended in erythrocyte lysis buffer (0.83% NH$_4$Cl in H$_2$O) for 8 min, passed through a 70 µm mesh and then spun at 1,200 g for 5 min. The pellet was washed once in 1× PBS, resuspended and passed through 40 µm mesh and spun at 1,200 g for 5 min. The pellet was resuspended in growth media (DMEM supplemented with 10% foetal bovine serum (FBS)). The SVP tubes that

remained on the 70 μm mesh were washed off and collected in growth media and cultured on microscope coverslips that were precoated with 1% fibronectin (Sigma: item no.: F1141). SVPs were imaged 12 h later for GFP occupancy[10]. For transfection experiments, expression vector harbouring cDNA for PDGFRβ was purchased from GE Dharmacon Openbiosystems (item no.: 30060666). Cells were reverse-transfected using Superfect (Qiagen: item no.: 301305) transfection regent following the manufacturer's protocol. Cell media (DMEM supplemented with 10% FBS) was replaced 12 h post-transfection.

**Cell culture treatments and qPCR.** *Cyclohexamide.* APCs were plated in 60 mm dishes. Twenty-four hours after plating, media was aspirated and cells were washed thrice with 1 × PBS. APCs were administered serum-free DMEM media and supplemented with 10 μg ml$^{-1}$ cyclohexamide (Sigma Aldrich: item no.: 01810) for 15 min before vehicle or rosiglitazone (Cayman Chemicals, item no.: 71740) treatment. mRNA was harvested 4 h after treatment.

*Rosiglitazone.* APCs were plated in six-well dishes. Twenty-four hours after plating, media was aspirated and cells were washed thrice with 1 × PBS. APCs were administered serum-free DMEM media supplemented with denoted concentrations of rosiglitazone (Cayman Chemicals, item no.: 71740). mRNA was harvested at the denoted times (4 and 48 h after treatment).

*PDGF-B and SU16F.* Cells were plated in six-well dishes. Twenty-four hours after plating, media was aspirated and cells were washed thrice with 1 × PBS. Cells were administered serum-free DMEM media supplemented with vehicle, PDGF-B (10 or 50 ng ml$^{-1}$; PeproTech item no.: 315-18) or SU16F (5 μM; Tocris Bioscience item no.: 3304). mRNA was harvested at the denoted times (4 and 48 h after treatment).

*Quantitative real-time PCR.* Total RNA was extracted using TRIzol (Invitrogen: item no.: 15596026) from either mouse tissues or cells. Mouse tissues ($n = 4$ individual tissues) and cells ($n = 4$ individual wells) were pooled and analysed in technical quadruplicates. These experiments were performed on three independent cohorts. cDNA synthesis was performed using RNA to the cDNA High Capacity Kit (Invitrogen: item no.: 4387406). Gene expression was analysed using Power SYBR Green PCR Master Mix with ABI 7500 Real-Time PCR System. qPCR values were normalized by 18S rRNA expression. Primer sequences are available in Supplementary Table 1.

**Flow cytometry and sorting.** SV cells were isolated as above and washed, centrifuged at 1,200g for 5 min and analysed with a FACScans analyser or sorted with a BD FACS Aria operated by the UT Southwestern Flow Cytometry Core (Supplementary Fig. 13). Data analysis was performed using the BD FACS Diva and FlowJo software. Sorting of GFP+ and GFP− cells was performed on live SV cells from P60 mice. For GFP+ (native fluorescence), CD31+ and SMA+ flow analysis, SV cells from AdipoTrak mice were stained with rat anti-CD31 (CD31; 1:200 BD Bioscience: item no.: 550274) and rabbit anti-α-smooth muscle actin (1:200, Sigma: item no.: SAB5500002) on ice for 30 min. For quantification of cell death, cells were stained with cleaved caspase-3 antibody (1:500, Cell Signaling Technology: item no.: 9661). For quantification of proliferation events, cells were stained with rat-anti-Brdu (1:100, Abcam: item no.: ab6326). Cells were then washed twice with the staining buffer and incubated with cy5 donkey anti-rat (1:500, Jackson ImmunoResearch, item no.: 711-605-152) and cy3 donkey anti-rabbit (1:500, Jackson ImmunoResearch, item no.: 711-605-150) secondary antibody for CD31 and SMA, respectively. Cells were incubated for 30 min on ice before flow cytometric analysis. For gating strategies of both GFP sorting and flow analysis, live cells were selected by size on the basis of FSC and SSC. Single cells were then gated on both SSC and FSC Width singlet's. SVF cells isolated from GFP-negative mice, along with primary-minus-one controls, were used as negative control to determine background fluorescence levels (Supplementary Figs 13 and 14).

**Migration assays.** *Wound-scratch assay.* FACS-isolated GFP APCs were plated in Ibidi chamber plates (item no.: 80206). Once confluent, the chamber was removed creating identical scratches under all conditions. Cells were washed thrice with 1 × PBS and photographed. DMEM supplemented with 5% FBS was added along with denoted ligands and migration was monitored and photographed 12 h later. Wound closure was quantitated using Image J (NIH) by measuring the distance between the leading edge of opposing sides. Quantitation was performed on images from three independent chambers from three independent mice (total of nine chambers per cohort)[61].

*Transwell migration chamber.* FACS-isolated GFP+ SV cells were plated in the upper chamber at a density of $5 \times 10^4$ and treated with the denoted ligands. Twelve hours later, the top of the migration screens (Corning) were cleaned and isolated and fixed in 4% paraformaldehyde and mounted on slides and imaged. Two images were taken from one chamber. The assay was performed on three individual chambers from three independent mice (total of nine chambers per cohort)[61].

**Adipose explant assay.** Subcutaneous IGW depots were excised and chopped into 1 mm by 3 mm pieces, using a Leica MZ8 stereo dissecting microscope. Each IGW depot can be cut into ∼15 explants. Immediately after isolation, explants were encased (completely covered) in growth factor-free Matrigel (BD Bioscience

item no.: 356231). Explants were cultured in DMEM media supplemented with 10% FBS and were continually treated with ligand(s) when required. Vascular outgrowths were photographed 5 days later using a Leica DMi8 or an Olympus BX41. Progenitor cell occupancy was measured by counting the number of GFP+ cells per capillary sprouts per image after merging the fluorescence and phase images. For all experiments, all explants were photographed from three individual mice per cohort performed thrice. Explant vessel sprout success rate in the control groups was ∼70%. Sprout length was measured using Image J. Branching point was classified as a sprout that deviated or branched from the main sprout[10].

**Chromatin immunoprecipitation assay.** AdipoTrak GFP+ APCs were FACS isolated from subcutaneous adipose depots from control mice and cultured for two passages. APCs were treated with vehicle or 1 μM rosiglitazone for 4 h. DNA was crosslinked with formaldehyde and sonicated into 200–1,000 bp fragments. Appropriate antibodies were used to immunoprecipitate pan-PPARγ (1:50 dilution) (Cell Signaling: item no.: 2430) and pan-RXR (1:10 dilution) (Santa Cruz: item no.: SC-774) and antibodies were attached to protein A/G agarose beads (Santa Cruz: item no.: SC-2003). DNA was precipitated using ethanol. PPREs were identified using the nubiscan nuclear receptor-binding site bioinformatics tool (http://www.nubiscan.unibas.ch/)[35,62]. qPCR primers were designed using primer design 3 (http://bioinfo.ut.ee/primer3-0.4.0/) and used to amplify the putative PPRE in the PDGFRβ (5′-CAGGCCTTTCATCACACTT-3′ (forward) and 5′-CCAAATGCTCTCCTCTCTGC-3′ (reverse)) and VEGF (5′-GGACCCTGGT AAGGGGTTTA-3′ (forward) and 5′-CAGCGGTGGAAGAAAAAGAG-3′ (reverse)). For the previously identified PPRE for Angptl4, we used the previously designed primers[36] (5′-TCTGGGTCTGCCCCCACTCCTGG-3′ (forward) and 5′-GTGTGTGTGTGGGATACGGCTAT-3′ (reverse).

**Gene expression microarray.** Adipose GFP+ APCs were sorted from six adult AdipoTrak-GFP control mice or Sox-PPARγ-LOF mice. These GFP+ SV cells were then subjected to gene expression microarrays performed on the Illumina Mouse-6 V2 BeadChip arrays by the UTSW Microarray Core[9] performed in triplicate. Genes considered having differential expression values between the control and mutant groups showed a more than twofold change in expression (at $P < 0.05$).

**Statistical analysis.** Statistical significance was assessed by two-tailed Student's *t*-test. Data are means and error bars are expressed as ± s.e.m. Mouse experiments were performed in biological triplicate with at least six mice per group and results are expressed as means ± s.e.m. No mice were excluded from the study unless visible fight wounds were observed. Cell culture experiments were collected from three or four independent cultures for each sample.

**Data availability.** The microarray data have been deposited in the Gene Expression Omnibus with an accession code GSE82328. The authors declare that all other data supporting the findings of this study are available within the article, its Supplementary Information or can be requested upon demand from the corresponding authors.

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

## Acknowledgements

We thank Dr Daochun Sun for bioinformatics assistance and heat map construction. This study was supported by the National Institute of Health and the National Institute of Diabetes and Digestive and Kidney Disease grants (R01 DK066556, R01 DK064261 and R01 DK088220) to J.M.G. D.C.B. is supported by the National Institute of Diabetes and Digestive and Kidney Disease grant K01 DK109027. Y.J. is supported by the National Institute of Diabetes and Digestive and Kidney Disease grant K01 DK111771.

## Author contributions

D.C.B., Y.J. and J.M.G. conceived, designed and analysed the experiments and wrote the manuscript. D.C.B., Y.J., A.J. and W.T. performed the experiments and analysed the data. R.W.A. and M.K. helped in the development of the TRE-PPARγ transgenic mouse line. All authors discussed the results and commented on the manuscript.

## Additional information

**Competing interests:** The authors declare no competing financial interests.

**Publisher's note**: 

