## [Peer Review File · Nature Communications]

Reviewer #1 (Remarks to the Author)

This paper shows that adipocyte precursor cells marked by PPAR γ expression regulate angiogenesis through the action of PPAR γ . Additionally, PPAR γ is required in precursors to regulate the association of precursor cells with their peri-vascular niche. Mechanistically, PPAR γ induces the expression of PDGFR β which plays an important role in promoting niche occupancy. Overall, this is an interesting and timely study that represents a substantial advance for the field. While the major conclusions are well supported, there are some important issues that should be addressed as indicated below:

Major issues:

1. The authors classify the adipose 'stem cells' solely via PPAR γ -tTA reporter activity driving expression of TRE-GFP, thus marking cells with activity at the PPAR γ promoter (even claiming that GFP-negative cells are 'not progenitors' in the text for figure 5). The authors use this same PPAR γ -tTA to drive Cre expression causing knockout of floxed-PPAR γ , then track the PPAR γ knockout cells using GFP, and observe a change in activity of these cells. Insufficient evidence is provided in this manuscript to conclude that PPAR γ promoter activity is a sufficient marker of all adipose stem cells, or that PPAR γ is the primary regulator of stem cell activity or niche occupancy, as is claimed in the text. The authors need to limit the scope of their conclusions to what is actually shown in their data (i.e. cells with active PPAR γ promoter activity are displaced from the vascular niche upon PPAR γ knockout). Given the significant possibility that GFP-negative stem cells may remain or reoccupy the vascular niche, staining for PPAR γ -independent, general stem cell markers may begin to address this issue. However, given the profound lipodystrophic phenotype induced by PPAR γ knockout, it will be difficult to definitively prove the physiologic relevance of either GFP+ or GFP- stem cells for adipogenesis.
2. One of the principle claims is the importance of PDGFR β for retention and activity of PPAR γ expressing cells in the perivascular location. However, the authors have relied exclusively on pharmacologic methods to reduce PDGFR β activity, including SU16F and imatinib which are not specific and have significant off-target effects. The authors should examine the requirement for PDGFR β via genetic manipulation.
3. Conflicting claims in Figure 7. Figure 7f shows that imatinib treated cells are confined along the vasculature which is contradictory to the previous claim that PDGFR β is required to retain GFP+ cells in the peri-vascular location. The authors claim that figure 7i shows displacement of GFP+ cells from the perivascular location however GFP+ cells are evident in proximity to the vessels in Figure 7i and the distance from vessel changes are small. Moreover, the GFP/SVP has profound error bars (in addition to the quantification issues discussed in minor issues).

More minor Issues:

1. Quantification of GFP+ cells on all SVP experiments should be per μM^2 of SVP because there may be differences in the size of SVP particles produced.
2. In-vitro vascular sprouting is highly variable between individual explants (a certain percentage of samples will never grow sprouts even in control conditions). The authors need to report how many explants were analyzed and ensure this number is statistically sufficient.
3. The authors should show at least one set of representative image of the GFP+ cell occupancy of the vascular sprouts quantified in (figure 2i, SF3k, 4f), which is as relevant to their major conclusions as angiogenesis measured by vascular sprout length.
4. In the SMA-Cre mice, deletion of PPAR γ from SMA+ mural cells causes no change to the vascular niche (the authors claim this is because of short 14 day time), but once again go on to show that the only effect of PPAR γ -knockout is on the PPAR γ -expressing cells, and there does not appear to be a large change in the number of PPAR γ -expressing cells in SF3f.
5. Typo Figure 5c is misreferenced in relation to rosiglitazone administration (should be 5d)

Reviewer #2 (Remarks to the Author)

In the manuscript entitled “A PPAR γ transcriptional cascade directs adipose progenitor-niche interaction and niche expansion” Jiang et al. show that PPAR γ , a master transcriptional regulator of adipogenesis, controls assembly and maintenance of the adipose tissue vascular niche via induction of vascular sprouting and recruitment of adipose precursors. They show that PPAR γ directly activates PDGFR β .

Although the findings are potentially interesting, there are concerns about novelty (see below). Moreover, this reviewer is not convinced that this ms is suited for the broad readership of Nature Communications.

Major comments:

1. Recent studies demonstrated that perivascular preadipocytes expressed PDGFR β ⁺ and contributed to both white and beige adipogenesis in inguinal WAT and were characterized by high Ppar1 and Ppar2 mRNA levels (Gupta et al., Cell Metab, 2012, Vishvanath et al., Cell Metab, 2016). The role of PDGFR β signaling in regulation of mural cells plasticity and adipose tissue development was shown in 2011 by Olson and Soriano (Developmental Cell).
2. The number of mice/biological replicates used for molecular, histological and phenotypic analyses is low and 3 mice per group is not enough. The authors should increase the number of biological replicates.
3. Figure 4 and Supplementary Figure 4 – Authors demonstrate that rosiglitazone treatment in mice increased the number of PECAM⁺ vessels. Pro-angiogenic effect of rosiglitazone in adipose tissue has already been demonstrated in humans and an involvement of PPAR γ has been suggested (Gealekman et al, 2012).
4. The authors should evaluate the mechanism of PPAR γ and PDGFR β signaling in more detail, especially downstream pathway(s). The role of PDGFR β signaling stimulating proliferation of vascular smooth muscle cells and mesenchymal cells has already been described (reviewed by Gaengel, 2009).
5. Figure 7 – Why is fat mass decreased when food intake is not changed/slightly increased in mice treated with imatinib? Why is fat mass presented as %body fat? The authors should measure and present both fat and lean mass content. Imatinib treated mice have smaller adipocytes – is just the size of adipocytes changed or also the phenotype? Is the expression of thermogenic markers increased? What about hepatic lipid levels?

Minor comments:

1. Page 9 – GW9662 is a PPAR γ antagonist NOT agonist.
2. The Figure legends in supplementary material have different font and some of them not described properly e.g. there is no description for Supplementary Figure 5e.
3. Figure 6b – it seems like pictures presented were taken at different magnifications.
4. Supplementary Figure 7 – For GTT analyses the authors should calculate AUC.

Reviewer #3 (Remarks to the Author)

In this study Jiang et al. provide very interesting findings about the role of PPAR γ expressing precursors in the regulation of adipose stem cell migration, niche vasculogenesis and angiogenesis, and stem cell occupancy of the niche. The author's data would imply that PPAR γ expression is required even before the precursors are established within in the perivascular region and that its expression is required to behave like mural cells in white adipose tissue vessels. This is novel and relevant in the context of identifying the cascade of events of adipose stem niche formation.

Specific points

1. From the images provided, it is not clear that PPAR γ LOF approaches lead to reduced number of blood vessels, and

PPAR γ GOF lead to increased number of blood vessels. The authors should provide more convincing analysis like performing whole mount-visualization of the vascular network in WAT depots.

2. It is not clear how the authors have assessed the expression of vascular markers by FACS. Is the total SVF analyzed? Not clear if the reduced expression of the vascular markers is due to reduce number of cells or reduced expression of the markers per se?

3. A critical question is how are the precursors (PPAR γ expressing cells) altering the angiogenic capacity of endothelial cells? Sprouting angiogenesis is the principal mechanism by which endothelial cells expand the vascular network in most of the tissue and this process is mediated by proliferation and migration of endothelial cells. In principal the PPAR γ expressing precursors do not become endothelial cells, so it is not clear how the reduced number of mural cells (GFP positive) is enough to have such a dramatic effect in terms of vessel growth. Stimulation of angiogenesis in the context of WAT is principally mediated by adipocytes, is it not possible that the reduced angiogenesis is due to reduced number of adipocytes differentiation?

4. Is the reduced vascularity in vivo leading to increased hypoxia?

5. Fig 3D implies that GFP positive cells (Precursors) disappear upon deleting PPAR γ . Do the authors have any explanation for this phenotype? So is PPAR γ expression required to attach to endothelial cells in the vessel wall?

Minor points

1. Some of the labeling in figures is not clear. For instance in Fig. 2b how was vessel quantification done? In Fig 2c “% Normalized flow” is not clear what it means. In Fig. 2D, 2E, 3F, 3J, 4C, 5B, 5C, 5D, 5E, 6A how was the expression analyzed? Compared to what? Normalized to what? All these labels should be clarified.

Reviewers' comments:

Reviewer #1 (Expert in adipose tissue biology; Remarks to the Author):

This paper shows that adipocyte precursor cells marked by PPAR γ expression regulate angiogenesis through the action of PPAR γ . Additionally, PPAR γ is required in precursors to regulate the association of precursor cells with their peri-vascular niche. Mechanistically, PPAR γ induces the expression of PDGFR β which plays an important role in promoting niche occupancy. Overall, this is an interesting and timely study that represents a substantial advance for the field. While the major conclusions are well supported, there are some important issues that should be addressed as indicated below:

We thank the reviewer for highlighting the importance of this work as well as their thoughtful comments that have helped bolster our scientific findings. We hope that we have adequately addressed the reviewers concerns and the manuscript warrants publication.

Major issues:

1. The authors classify the adipose 'stem cells' solely via PPAR γ -tTA reporter activity driving expression of TRE-GFP, thus marking cells with activity at the PPAR γ promoter (even claiming that GFP-negative cells are 'not progenitors' in the text for figure 5). The authors use this same PPAR γ -tTA to drive Cre expression causing knockout of floxed-PPAR γ , then track the PPAR γ knockout cells using GFP, and observe a change in activity of these cells. Insufficient evidence is provided in this manuscript to conclude that PPAR γ promoter activity is a sufficient marker of all adipose stem cells, or that PPAR γ is the primary regulator of stem cell activity or niche occupancy, as is claimed in the text. The authors need to limit the scope of their conclusions to what is actually shown in their data (i.e. cells with active PPAR γ promoter activity are displaced from the vascular niche upon PPAR γ knockout). Given the significant possibility that GFP-negative stem cells may remain or reoccupy the vascular niche, staining for PPAR γ -independent, general stem cell markers may begin to address this issue. However, given the profound lipodystrophic phenotype induced by PPAR γ knockout, it will be difficult to definitively prove the physiologic relevance of either GFP+ or GFP- stem cells for adipogenesis.

The reviewer is correct that we should accurately limit our conclusions to the data presented. We have altered the text to reflect this. We, however, must disagree with the reviewer's statements that AdipoTrak does not accurately represent all adipose stem/progenitor cells. We have detailed these findings in other published manuscripts (Jiang et al, Cell Reports 2014 and Tang et al; Science 2008) demonstrating that no other cell type (AdipoTrak negative; GFP-) can compensate for the loss of AdipoTrak+ cells under our conditions and settings. Thus these data indicated that AdipoTrak marks essentially the entire adipose stem/progenitor lineage. We would agree with the reviewer that PPAR γ might not be always expressed at a given moment in a marked AdipoTrak cell given the nature of the tracking system. As well as the notion that depending on the lineage commitment of a given APC that not all AdipoTrak stem/progenitor cells are equal. That is some cells may be more committed to the adipose lineage than others and we are currently working on ways and new genetic tools to examine this. We definitely appreciate any feedback the reviewer may have in identifying such avenues.

2. One of the principle claims is the importance of PDGFR β for retention and activity of PPAR γ expressing cells in the perivascular location. However, the authors have relied exclusively on pharmacologic methods to reduce PDGFR β activity, including SU16F and imatinib which are not specific and have significant off-target effects. The authors should examine the requirement for PDGFR β via genetic manipulation.

We have added the genetic deletion of PDGFR β as well as a PDGFR β consecutively active genetic model. Please see Figures 4, 5, and 6.

3. Conflicting claims in Figure 7. Figure 7f shows that imatinib treated cells are confined along the vasculature which is contradictory to the previous claim that PDGFR β is required to retain GFP+ cells in the peri-vascular location. The authors claim that figure 7i shows displacement of GFP+ cells from the perivascular location however GFP+ cells are evident in proximity to the vessels in Figure 7i and the distance from vessel changes are small. Moreover, the GFP/SVP has profound error bars (in addition to the quantification issues discussed in minor issues).

We apologize for the confusion. We have repeated the experiments on AT-GFP control mice to specifically examine AT-APC marked progenitors. We have changed the figure to more accurately depict the AdipoTrak labeled cells. We have taken new higher magnification images to examine if APCs are displaced from the vasculature. We have quantified more SVPs to reduce the statistical error, as suggested and have been noted in the methods.

More minor Issues:

1. Quantification of GFP+ cells on all SVP experiments should be per μM^2 of SVP because there may be differences in the size of SVP particles produced.

We have made these changes (GFP #/100 μm SVP). We also evaluated more SVPs to improve statistical error as suggested.

2. In-vitro vascular sprouting is highly variable between individual explants (a certain percentage of samples will never grow sprouts even in control conditions). The authors need to report how many explants were analyzed and ensure this number is statistically sufficient.

We agree that explant sprouting is highly variable. In the methods section, we have reported the number of explants and images analyzed. We have also included a statement about the infrequency of sprouting (30% of control explants fail to sprout under our conditions)

3. The authors should show at least one set of representative image of the GFP+ cell occupancy of the vascular sprouts quantified in (figure 2i, SF3k, 4f), which is as relevant to their major conclusions as angiogenesis measured by vascular sprout length.

We have added images identifying GFP+ cells along newly sprouted vessels (Supplementary Fig. 7n)

4. In the SMA-Cre mice, deletion of PPAR γ from SMA+ mural cells causes no change to the vascular niche (the authors claim this is because of short 14 day time), but once again go on to show that the only effect of PPAR γ -knockout is on the PPAR γ -expressing cells, and there does not appear to be a large change in the number of PPAR γ -expressing cells in SF3f.

We agree there is no major changes in regards to adipose tissue mass or adipose tissue niche markers or changes in PPAR γ expressing cell number. The major change we observe is their displacement away from the blood vessel niche. We observe changes in the angiogenic genetic signature of these PPAR γ expressing which we believe is due to the loss of niche occupancy.

5. Typo Figure 5c is misreferenced in relation to rosiglitazone

administration (should be 5d).

Thank you! This has been corrected.

Reviewer #2 (Expert in adipose tissue biology; Remarks to the Author):

In the manuscript entitled "A PPAR γ transcriptional cascade directs adipose progenitor-niche interaction and niche expansion" Jiang et al. show that PPAR γ , a master transcriptional regulator of adipogenesis, controls assembly and maintenance of the adipose tissue vascular niche via induction of vascular sprouting and recruitment of adipose precursors. They show that PPAR γ directly activates PDGFR β .

Although the findings are potentially interesting, there are concerns about novelty (see below). Moreover, this reviewer is not convinced that this ms is suited for the broad readership of Nature Communications.

We appreciate the reviewer's critical assessment of our manuscript and have taken into consideration his or hers comments. We believe that these comments have helped strengthen our conclusions and have helped shaped our future research direction. We hope the adjusted manuscript is now suitable for publication.

Major comments:

1. Recent studies demonstrated that perivascular preadipocytes expressed PDGFR β ⁺ and contributed to both white and beige adipogenesis in inguinal WAT and were characterized by high Ppar γ 1 and Ppar γ 2 mRNA levels (Gupta et al., Cell Metab, 2012, Vishvanath et al., Cell Metab, 2016). The role of PDGFR β signaling in regulation of mural cells plasticity and adipose tissue development was shown in 2011 by Olson and Soriano (Developmental Cell).

We do not disagree with the reviewer and we are not questioning the involvement of PDGFR β ⁺ cells in the generation of new adipocytes as we and others (Drs. Gupta and Soriano) have discovered. We are expanding on this notion, which suggests that PDGFR β has a functional role in adipose progenitor cell biology. We found that PDGFR β regulates adipose progenitor cell-niche interaction and niche and vascular expansion. We apologize if this was not clear and have emphasized this more in the manuscript.

2. The number of mice/biological replicates used for molecular, histological and phenotypic analyses is low and 3 mice per group is

not enough. The authors should increase the number of biological replicates.

We have incorporated more mice/study. The numbers are now 10 mice/group replicated 3 times.

3. Figure 4 and Supplementary Figure 4 - Authors demonstrate that rosiglitazone treatment in mice increased the number of PECAM+ vessels. Pro-angiogenic effect of rosiglitazone in adipose tissue has already been demonstrated in humans and an involvement of PPAR γ has been suggested (Gealekman et al, 2012).

We do not disagree with the reviewer and we are not questioning the involvement of rosiglitazones' ability to expand the vascular niche. Rather, we are expanding on this notion to define a cellular and molecular mechanism for why transcriptional activation of PPAR γ causes adipose tissue vascular remodeling. We apologize if this was not clear and have emphasized this more in the manuscript.

4. The authors should evaluate the mechanism of PPAR γ and PDGFR β signaling in more detail, especially downstream pathway(s). The role of PDGFR β signaling stimulating proliferation of vascular smooth muscle cells and mesenchymal cells has already been described (reviewed by Gaengel, 2009).

These efforts are ongoing. But as recommended by reviewer 1 we have added a consecutively active PDGFR β genetic model to the manuscript demonstrating robust adipose progenitor cells niche interaction. In the future, we hope this model will help identify the signaling pathway that is required for progenitor cell-niche interaction.

5. Figure 7 - Why is fat mass decreased when food intake is not changed/slightly increased in mice treated with imatinib? Why is fat mass presented as %body fat? The authors should measure and present both fat and lean mass content. Imatinib treated mice have smaller adipocytes - is just the size of adipocytes changed or also the phenotype? Is the expression of thermogenic markers increased? What about hepatic lipid levels?

We have added the lean mass data, which shows a slight increase. We believe that the reviewers' comments are beyond the scope of work. We definitely agree with the reviewer that these are interesting findings, which we are actively pursuing but are not fully understood at this point in time.

Minor comments:

1. Page 9 - GW9662 is a PPAR γ antagonist NOT agonist.

We apologize for the typo and this has been corrected.

2. The Figure legends in supplementary material have different font and some of them not described properly e.g. there is no description for Supplementary Figure 5e.

This has been corrected.

3. Figure 6b - it seems like pictures presented were taken at different magnifications.

This has been corrected

4. Supplementary Figure 7 - For GTT analyses the authors should calculate AUC.

AUC has been calculated.

Reviewer #3 (Expert in angiogenesis; Remarks to the Author):

In this study Jiang et al. provide very interesting findings about the role of PPAR γ expressing precursors in the regulation of adipose stem cell migration, niche vasculogenesis and angiogenesis, and stem cell occupancy of the niche. The author's data would imply that PPAR γ expression is required even before the precursors are established within in the perivascular region and that its expression is required to behave like mural cells in white adipose tissue vessels. This is novel and relevant in the context of identifying the cascade of events of adipose stem niche formation.

We thank the reviewer for noting that are work is novel and relevant in the context of adipose stem cell-niche biology. We also thank the reviewer for their thoughtful comments that have expanded our findings on the possible role of adipose progenitor cells and vasculature expansion.

Specific points

1. From the images provided, it is not clear that PPAR γ LOF approaches lead to reduced number of blood vessels, and PPAR γ GOF lead to increased number of blood vessels. The authors should provide more convincing analysis like performing whole mount-visualization of the vascular network in WAT depots.

We have provided CD31 staining of adipose tissue sections from our PPAR γ loss and gain of function models to elevate blood vessel number (Fig. 2a). Thank you for the recommendation.

2. It is not clear how the authors have assessed the expression of vascular markers by FACS. Is the total SVF analyzed? Not clear if the reduced expression of the vascular markers is due to reduce number of cells or reduced expression of the markers per se?

We apologize for the confusion and have clarified the text, figures and the methods section. We have analyzed both cell number and gene and protein expression of vascular markers.

3. A critical question is how are the precursors (PPAR γ expressing cells) altering the angiogenic capacity of endothelial cells? Sprouting angiogenesis is the principal mechanism by which endothelial cells expand the vascular network in most of the tissue and this process is mediated by proliferation and migration of endothelial cells. In principal the PPAR γ expressing precursors do not become endothelial cells, so it is not clear how the reduced number of mural cells (GFP positive) is enough to have such a dramatic effect in terms of vessel growth. Stimulation of angiogenesis in the context of WAT is principally mediated by adipocytes, is it not possible that the reduced angiogenesis is due to reduced number of adipocytes differentiation?

Towards this end, we have identified that PPAR γ transcriptionally targets VEGF within APCs. VEGF is then secreted and then targets endothelial cell growth (Figures 7-9 and Supplementary Figs 10-12). As noted in the discussion, we believe that the presence of APCs along the vascular signal to the adipose tissue vascular to expand and continue making new adipocytes however, if these cells are displaced from their niche vascular expansion is halted and adipogenesis is compromised.

4. Is the reduced vascularity in vivo leading to increased hypoxia?

This is a very important question that we plan to examine in the future. We also know that other research groups are currently examining hypoxia in adipose tissues thus we don't want to overlap.

5. Fig 3D implies that GFP positive cells (Precursors) disappear upon deleting PPAR γ . Do the authors have any explanation for this phenotype? So is PPAR γ expression required to attach to endothelial cells in the vessel wall?

We don't believe the cells necessarily disappear but rather are located next to the vasculature. We have not followed these cells long enough to track their fate.

Yes, we proposed in figures 1-7 that PPAR γ via PDGFR β regulate the ability adipose progenitor cells to interact with the blood vessel. Without either PPAR γ or PDGFR β there is reduced progenitor cell presence at the blood vessel wall reducing overall adipogenic potential.

Minor points

1. Some of the labeling in figures is not clear. For instance in Fig. 2b how was vessel quantification done? In Fig 2c "% Normalized flow" is not clear what it means. In Fig. 2D, 2E, 3F, 3J, 4C, 5B, 5C, 5D, 5E, 6A how was the expression analyzed? Compared to what? Normalized to what? All these labels should be clarified.

Labels have been clarified

Reviewers' Comments:

Reviewer #1:

Remarks to the Author:

The manuscript has been substantially improved. In particular the inclusion of genetic manipulations to examine the role of PDGFRb signaling strengthens the study. The paper presents important and interesting results that advance the field. I also appreciate that the authors altered the text, specifically changing most references from 'stem cell' to 'adipocyte precursor cell' (APC), which more accurately reflects the physiological properties of the PPAR γ marked perivascular cells.

The underlying premise of this manuscript is predicated upon the use of PPAR γ (an obligatory master regulator of adipogenesis that must be expressed by all preadipocytes at some point in differentiation), as a marker for what they have identified as 'stem cell-like' progenitors. In their response, the authors argue that PPAR γ expression marks the 'entire' stem cell/precursor/niche population, using evidence that deletion of PPAR γ blocks adipogenesis. However, the validity of this argument is diminished by the fact that PPAR γ expression is required for differentiation of preadipocytes, so the authors cannot conclude if PPAR γ knockout is disrupting precursor function, or simply blocking differentiation.

Reviewer #2:

Remarks to the Author:

Important questions of Rev 2 and 3 were not addressed ("We believe that the reviewers' comments are beyond the scope of work. We definitely agree with the reviewer that these are interesting findings, which we are actively pursuing but are not fully understood at this point in time."; "4.

Is the reduced vascularity in vivo leading to increased hypoxia?

This is a very important question that we plan to examine in the future. We also know that other research groups are currently examining hypoxia in adipose tissues thus we don't want to overlap.")

So I stick with my initial assessment: better suited for another more specialized journal

Reviewer #3:

Remarks to the Author:

Overall the manuscript has largely improved with additions and clarifications. However I still have minor concerns.

1. The manuscript requires simplification in the content and in the message.
2. How does PDGFB/PDGFRB regulate the angiogenic response? By promoting the expression of angiogenic factors per se or by regulating APC residency and that in turn stimulating the expression of angiogenic factors. This concept should be at least better discussed.
3. Would the observation that VEGF expression in APCs is dependent on PPAR γ imply that it is independent of hypoxia? How would that fit within the canonical way of activating VEGF?

REVIEWERS' COMMENTS:

Reviewer #1 (Remarks to the Author):

The manuscript has been substantially improved. In particular the inclusion of genetic manipulations to examine the role of PDGFRb signaling strengthens the study. The paper presents important and interesting results that advance the field. I also appreciate that the authors altered the text, specifically changing most references from 'stem cell' to 'adipocyte precursor cell' (APC), which more accurately reflects the physiological properties of the PPARg marked perivascular cells.

Thank you!

The underlying premise of this manuscript is predicated upon the use of PPARg (an obligatory master regulator of adipogenesis that must be expressed by all preadipocytes at some point in differentiation), as a marker for what they have identified as 'stem cell-like' progenitors. In their response, the authors argue that PPARg expression marks the 'entire' stem cell/precursor/niche population, using evidence that deletion of PPARg blocks adipogenesis. However, the validity of this argument is diminished by the fact that PPARg expression is required for differentiation of preadipocytes, so the authors cannot conclude if PPARg knockout is disrupting precursor function, or simply blocking differentiation.

We have added appropriate wording detailing the limitations of the experimental methods and models surrounding PPARg gain and loss of function models.

Reviewer #2 (Remarks to the Author):

Important questions of Rev 2 and 3 were not addressed ("We believe that the reviewers' comments are beyond the scope of work. We definitely agree with the reviewer that these are interesting findings, which we are actively pursuing but are not fully understood at this point in time."; "4. Is the reduced vascularity in vivo leading to increased hypoxia?

This is a very important question that we plan to examine in the future. We also know that other research groups are currently examining hypoxia in adipose tissues thus we don't want to overlap.")

So I stick with my initial assessment: better suited for another more specialized journal

We apologize that we were unable to satisfy reviewer 2's comments

Reviewer #3 (Remarks to the Author):

Overall the manuscript has largely improved with additions and clarifications. However I still have minor concerns.

1. The manuscript requires simplification in the content and in the message.

We believe the updated manuscript has been clarified and simplified. Thank you!

2. How does PDGFB/PDGFRB regulate the angiogenic response? By promoting the expression of angiogenic factors per se or by regulating APC residency and that in turn stimulating the expression of angiogenic factors. This concept should be at least better discussed.

We agree that this concept is important and has been included in the revised manuscript. We believe that both progenitor cell niche interaction and vascular niche expansion are coupled. Thus when one is disrupted or enhanced so is the other. Ongoing studies in the lab focus on how PDGFRb mediates these events and whether PDGFRb induces the expression of angiogenic genes and/or cues.

3. Would the observation that VEGF expression in APCs is dependent on PPAR γ imply that it is independent of hypoxia? How would that fit within the canonical way of activating VEGF?

We cannot rule out hypoxia as an inducer of VEGF in APCs since we did not directly test this. What we can conclude is that VEGF expression is induced by PPAR γ transcriptional activation; however, how these two pathways interplay should be further evaluated in future research. We have added a discussion point around this critical question. Thank you for highlighting this.